



# Seasonal and interannual variability of sea-ice state variables: Observations and predictions for landfast ice in northern Alaska and Svalbard

Marc Oggier[1], Hajo Eicken[1], Meibing Jin[1], Knut Høyland[2,3]

[1]: International Arctic Research Center, University of Alaska Fairbanks, PO Box 757340, Fairbanks, AK 99775, USA

2: Norwegian University of Science and Technology, Høgskoleringen 7a, NO-7491 Trondheim, Norway,

3: The University Centre in Svalbard, P.O. Box 156, N-9171 Longyearbyen, Norway

*Correspondence to*: Marc Oggier (moggier@alaska.edu)

**Abstract.** Validation of sea-ice models, representation of sea-ice processes in large-scale models, and regional planning around ice use and hazards requires climatological ice property data. We summarize key ice properties, in particular temperature and salinity, representative of broader Arctic conditions, from long-
term observations near Utqiaġvik, Alaska and Van Mijen Fjord, Svalbard. Additionally, we simulate salinity and temperature profiles using the Los Alamos sea-ice model (CICE) in stand-alone mode, forced with meteorological data for both locations. We compare observations and model results by aggregating profiles using a degree day model and statistical analysis to create ice property climatologies, which describe the seasonal evolution of sea ice. During the growth season, the CICE model accurately replicates ice property
evolution for both salinity (R=0.7) and temperature (R=0.9). While the model initiates ice desalination at melt onset, and reproduces the temperature field well through melt (R=0.9), model salinities later tend towards an asymptotic value of 5‰ (R=0.3). This suggests that the model does not fully capture the desalination processes and their impact on ice physico-chemical properties during the melt season. Overall, the standard deviation of the model remains similar to the natural sea-ice variability throughout the season. Despite mismatches during
the melt season, the CICE model shows promise for simulating the seasonal evolution of salinity and temperature profiles, which may serve as proxies for bulk ice properties that constrain transport of heat and mass through sea ice. Our findings highlight the necessity for a large number of observations throughout the year to create an effective model benchmarking dataset.



## 1. Introduction

Sea ice controls the interaction between ocean and atmosphere and is a key aspect of polar marine ecosystems (Cottier et al., 1999). Heat transport and light transmission across the ocean-ice-air interface are controlled by sea-ice properties, ice microstructure, and stratigraphy. Ice transport properties also govern the sea-ice role as a habitat for a broad range of microscopic organisms thriving in the bottommost ice layer, which in turn supports zooplankton grazers and higher trophic levels (Bluhm and Gradinger, 2008; Gradinger et al.,

2010; Horner et al., 1992). Increasing industrial and maritime activities in the Arctic have the potential to stress the ecosystem. The risk of oil spills is of particular concern, with increasing ship traffic and continued interest in offshore hydrocarbon development (Brigham, 2010; Eguíluz et al., 2016; Smith and Stephenson, 2013). Information about the state of the sea-ice cover, and key sea-ice properties relevant to transport processes and entrainment of pollutants is critical in this context as well. In addition, compilation and synthesis of existing

datasets and model simulations can help provide important baseline data for a range of studies concerned with air-ocean interaction, and the ecological significance of sea ice.

Sea ice is a composite material, with brine trapped within the ice matrix during sea-ice growth. Under quiescent ice growth conditions typical of the Arctic, a high-porosity skeletal layer of few mm to cm thickness with fully connected mm- to sub-mm pore space makes up the bottommost ice (Petrich and Eicken, 2017). In growing

sea ice, brine channels several cm to dm in vertical extent develop as a result of convective overturning and drainage processes (Cole and Shapiro, 1998; Wells et al., 2011). The brine-filled pore space responds to cooling or warming through increases or decreases, respectively, in the brine volume fraction. These changes in turn affect pore microstructure and ice permeability, and impact macro-scale behaviour (Petrich and Eicken, 2017).

Assur (1960) and Zubov (1963) summarized years of field observations of pioneering physical and chemical sea-ice property measurements in the Arctic. Thorough analysis of sea ice led to the development of semi-empirical equations describing physical property evolution (Cox and Weeks, 1983, 1986; Leppäranta and Manninen, 1988). In addition, studies of the temporal evolution and spatial variability of sea-ice salinity and property profiles in the Arctic (e.g., Cox & Weeks, 1988a; Nakawo & Sinha, 1981) and Antarctic (Eicken, 1992;

Gough et al., 2012) provided further insights into processes impacting the vertical profile of sea-ice properties. This work has been expanded in recent years through the application of mush layer theory (Feltham et al., 2006; Notz and Worster, 2009) and the development of in-situ measurement approaches based on the change of sea-ice dielectric property, related to the change of pore fraction and microstructure, which in turn influences key sea-ice transport properties (Jones et al., 2012; O'Sadnick et al., 2016).

Driven by the need to improve representation of sea-ice processes in general circulation and Earth system models, more sophisticated sea-ice models were developed, aiming in particular to represent the seasonal sea-ice cycle more accurately (Griewank and Notz, 2015; Hunke et al., 2013). The availability of ice property data that captures the full range of variability is essential for validation of such modelling efforts. At the same time, mean and variance of key ice properties as a function of ice depth are essential in constraining the

analysis and simulation of biogeochemical or contaminant transport processes in sea ice (Oggier et al., 2019; Steiner et al., 2016). However, obtaining sea-ice property data representative of the temporal and spatial





variability are challenging due to the episodic nature of typical field measurement programs, the remoteness of field sites, and the lack of a consistent sampling protocol. Sustained observations of sea-ice property evolution at a single site, where ice conditions are comparable from year to year, are scarce.

In this study, we assembled ice properties (temperature, salinity, ice thickness) and meteorological data for two landfast ice sites representative of broader Arctic conditions: Utqiaġvik (formerly Barrow), Alaska, and Van Mijen Fjord, Svalbard. From these data, we derived an ice property climatology along with an evaluation of seasonal and interannual variability. Drawing on this data, we assess the ability of a standard Earth system model to replicate key aspects of the seasonal cycle as well as interannual variability. We chose the Los

Alamos sea-ice model (CICE; Hunke et al., 2013) for its capability to be run both in fully coupled atmosphere-land global climate models and as a standalone model. We evaluated the variability from observations and model results, with a focus on properties such as porosity and permeability, which are relevant for biogeochemical processes (Miller et al., 2015) and contaminant transport (Maus et al., 2015; Petrich et al., 2006).

**2.    Methods**

We analyzed ice core data from two different locations, representative of a broader range of quiescent growth conditions of Arctic first-year sea ice: Utqiaġvik, located at the northern tip of Alaska and Van Mijen Fjord on the west coast of the Svalbard Archipelago, 60 km south of Longyearbyen. At each location, field measurements and ice sampling were part of sustained ice observation programs (Druckenmiller et al., 2009;

Ervik et al., 2014; Høyland, 2009). In this study, we consider data collected during the ice seasons from 1997/98 through 2017/18.

**2.1.    Study sites**

2.1.1. **Utqiaġvik, Alaska**

The field sites were located 2 to 5 km southwest of Point Barrow, in a location protected from alongshore ice

movement and deformation, resulting in a level landfast ice cover growing undisturbed throughout the entire length of the landfast ice season (Figure 1–a). Data were derived from ice cores and *in situ* measurements of ice and snow thickness, and temperature profiles through the depth of the snow and ice cover at a mass balance site (MBS) installed from the 2006/07 through the 2015/16 ice seasons (Druckenmiller et al., 2009). While the position of the station varied slightly from year to year, the sites were representative of level,

undeformed, landfast sea ice formed under calm conditions (Figure 1). Sampling dates were scheduled to capture and follow the fundamental changes in sea-ice state throughout the winter and spring: young first-year sea ice in January, mature sea ice from April until mid-May, and desalination and melt through mid-June. Station locations are shown in Figure 1—a. Our analysis includes 180 ice cores, with 106 cores collected during 42 growth season sampling events, and 74 ice cores for a total of 41 melt season sampling events. The

National Oceanic and Atmospheric Administration (NOAA) records meteorological data at Will Rogers Memorial Airport station (PABR) in Utqiaġvik. The airport is located approximately 10 km southwest of the MBS location. Temperatures between the airport and the coring site are well correlated (Figure 2).





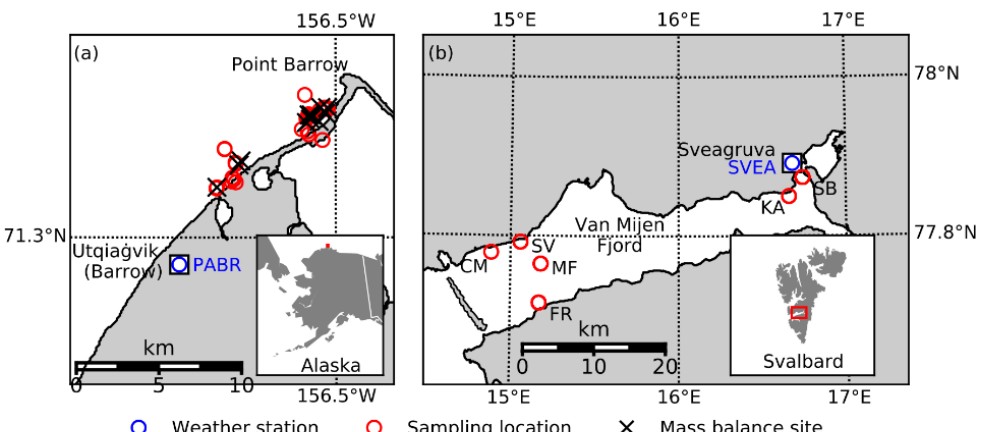

**Figure 1—Location of the two field sites with each core sampling location represented as by a red circle. (a) in Utqiaġvik, the position of the mass balance site (black crosses) and core sampling locations (red circles) vary slightly from 2006 to 2018, but remained protected by Point Barrow. The PABR weather station (blue circle) is located at local Will Rogers Memorial airport in Utqiaġvik. (b) in Van Mijen Fjord, sampling sites are spread across Van Mijen Fjord, and the SVEA weather station is located at Sveagruva**

### 2.1.2. Van Mijen Fjord, Svalbard

Ice cores and in-situ temperature measurements were collected at different locations (Figure 1—b) in Van Mijen Fjord during the winter 1998/99 through 2004/05, and in 2006/07, 2013/14 (Ervik et al., 2014; Høyland, 2009; Høyland et al., 2020) and 2016. An island partially blocks the mouth of the fjord and currents are mostly dominated by tides. In addition, glacial freshwater input from the surrounding mountains lowers the salinity of the fjord water (Høyland, 2009). Located on the west coast of Svalbard Archipelago, warm Atlantic water from the West Spitsbergen Current also influences ice growth conditions in Van Mijen Fjord (Gerland & Hall, 2006). Warm spells with air temperatures above 0 °C are not unusual before the onset of melt (Norwegian Meteorological Institute, 2016). The ice growth season was overall shorter and warmer than at Utqiaġvik. Landfast sea-ice is representative of undeformed ice grown under calm, mild conditions. The fjord is often divided into an outer and inner basin (Kangas, 2000). Høyland (2009) observed an ice thickness difference of 0.1 m between the two regions in 2004 and attributed the thinner ice in the outer basin to a later freeze-up and higher ocean heat flux. While the exact location of coring sites varies between sampling events and from year to year, for the purpose of this study we aggregated all ice cores into a dataset representative of the area. Figure 1 shows the field sampling sites. Our study included 60 ice cores, with 35 cores collected during 22 growth season sampling events, and 25 cores collected during 14 melt season sampling events. Meteorological data, from the Sveagruva Airfield on the north shore of the fjord, 30 km away from its mouth, were provided by the Norwegian Meteorological Institute.

### 2.2. Field measurements and sample/data processing

#### 2.2.1. Ice sampling

Similar sampling protocols were used in Utqiaġvik and Van Mijen Fjord. Ice cores were obtained with a fiberglass barrel ice corer (9 cm diameter at Utqiaġvik and 7 cm diameter in Van Mijen Fjord; for details see





Eicken et al., 2014). Snow depth, ice thickness, freeboard, and air, snow/air, snow/ice temperatures were measured. Temperature profiles were measured by inserting a thermistor probe into holes drilled at 5 or 10 cm intervals in an ice core immediately after extraction. Measurement precision and accuracy were in the range

of 0.05-0.1 °C and 0.1-0.3°C, respectively. Parallel cores were cut into 5 cm (occasionally 2.5 or 10 cm) thick horizontal slices after extraction and transferred to a sealed container for melting at room temperature. Bulk salinity was measured with a conductivity probe. The YSI 30 probe (YSI Incorporated, Yellow Springs, OH, USA) used in Utqiaġvik has a precision and an accuracy ranging from 0.05 to 0.1‰ and 0.1 to 0.2‰, respectively.

In-situ thermistor string measurements (10 cm vertical spacing, EBA Engineering, Edmonton, Canada) were available from Van Mijen Fjord for the years 1998 to 2004 to supplement ice core temperature measurements. In Utqiaġvik, temperature profiles recorded with an in-situ thermistor array (10 cm vertical resolution) at the MBS from 2006 through 2016 were used to supplement the ice-core measurements. The MBS was an automated station installed annually providing 15-or-30-minute interval measurements of ice temperature,

thickness information for ice and snow, sea-level measurements, temperature and relative humidity of the local atmosphere (2 m above sea-level) (Druckenmiller et al., 2009).

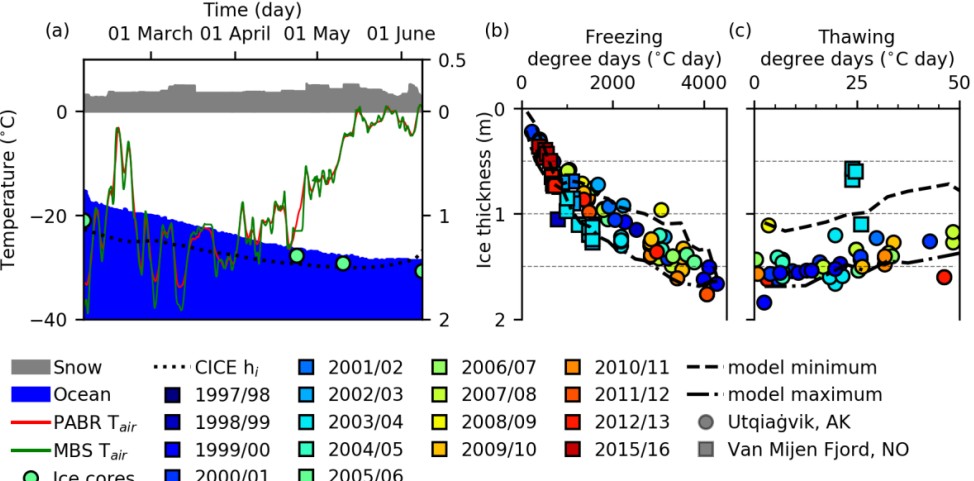

**Figure 2—Typical growth of level first-year ice: (a), sea ice (white) and snow (gray) thickness measured at the mass balance site in Utqiaġvik during the winter 2005/06 relative to the ice thickness measured in core holes during field work (green dots). The dotted line represents the simulated ice growth using the CICE model and weather data, including air temperatures collected at the PABR airport meteorological station (red line) or at the MBS (green line). Ice thickness measured in core holes as a function of degree days during the growth season (b) and melt season (c). The dashed lines correspond to the minimal and maximal ice thickness by 100 freezing degree days or 5 thawing degree days during the growth or melt, respectively, as simulated by the CICE model for Utqiaġvik between 1998 and 2014. Note that the maximum number value of FDD in Van Mijen Fjord is about 2000 FDD, while it reaches 4500 FDD in Utqiaġvik.**

### 2.2.2. Ice Core Degree Day Classification

While two thirds of cores were cut into 5 cm sections, segments of different lengths due to core breaks or intentional sampling of 10 cm or 2.5 cm segments need to be taken into consideration. We first resampled

core property depth profiles into 5 cm vertical sections with the zero-depth reference set at the ice surface. We





used a weighted average interpolation method, with the weight given by the overlapping length between the resampled section and each of the original sections. To account for ice thickness variations between cores belonging to the same class, we then resampled the ice cores using the same method, but setting the zero-depth reference at the bottom. Aligning ice cores at the ice-ocean interface reduced the variation in the data when analyzing the lower part of the ice cover.

Since ice thickness, salinity and temperature evolved slowly during the growth season, cores extracted over a span of a few days were assigned to the same coring event. During the melt season, only single-day sampling events were considered.

We relied on a degree day (DD) approach to describe the growth or melt stages of sea ice for individual core sampling events to quantify the temporal sea-ice evolution and to facilitate comparison between different years, seasons and locations. We used the daily average air temperature, $T_a$, and defined the freezing, respectively thawing, degree days as the cumulative difference using the freezing point of sea water as reference temperature, $T_{F, SW} = -1.8$ °C.

$$FDD = \sum_{t_0}^{t} min\big(0, T_a - T_{f,sw}\big) < 0 \text{ DD (°C day)} \tag{1}$$

$$TDD = \sum_{t_0}^{t} max\big(0, T_a - T_{f,sw}\big) > 0 \text{ DD (°C day)} \tag{2}$$

Freezing degree days (FDD) calculated from the freeze-up-day to the coring date of a given core provided a measure of both the duration and severity of cooling during the growth season. For Utqiaġvik, we used the locally observed freeze-up date to compute DD for the observations, while we used the freeze-up date derived from the CICE model for the simulation, defined as the first day of freezing in the uppermost layer without subsequent full melting until summer. For Van Mijen Fjord, in absence of local observation, we used the freeze-up date derived from the CICE model for both observation and simulations. Similarly, thawing degree days (TDD) indicated the extent of warming that the ice cover and the core extracted from it may have experienced. The initial time, $t_0$, to compute TDD was set to May 1st for Utqiaġvik and April 1st for Van Mijen Fjord to exclude brief warm spells before the onset of ice melt. Finally, we grouped the ice cores in intervals from 0, −300, −600, −1000, −1500 … −4500 DD (°C day) during the growth season and 0, 5, 10, 15, 25, 50, 100 DD (°C day) during the melt. For each DD interval, we computed summary statistics (mean, standard deviation, minimum and maximum) for ice properties.

### 2.3.    Model Simulations

We used the Los Alamos Sea Ice (CICE) model (Hunke et al., 2013) to simulate ice growth/melt and the seasonal changes of the ice internal temperature and salinity profiles. The CICE model, designed for fully coupled global climate models, was run in standalone mode, with an idealized ocean mixed layer under sea ice. The CICE model employed a two-mode brine gravity drainage formulation based on mushy-layer theory and shown to simulate the bulk salinity with sufficient accuracy throughout the seasonal cycle (Turner et al., 2013; Turner and Hunke, 2015). Hindcast simulations were forced with 6-hr National Center for Environmental Prediction (NCEP) reanalysis data combined with available local observational data. The CICE model was run on a single grid cell with 20 ice layers distributed across the ice thickness. The high vertical resolution provides a good representation of the steep salinity and temperature gradients near the top and bottom ice surface (Hunke, 2014). The model time step is 1 hour. To capture the full seasonal cycle a non-zero ocean-to-ice heat



flux was introduced during the summer months to represent the horizontal advection of heat to the site. The
magnitude of this flux was tuned based on comparison with observed ice thicknesses and snow depths.

For Utqiaġvik, meteorological data were obtained from the National Climate Data Center available for Wiley
Post-Will Rogers Memorial Airport (PABR), except for precipitation and humidity, which were provided by the
NCEP reanalysis model. Simulations were run for the period 1979 to 2013. In Van Mijen Fjord, forcing data
are from 6-hourly NCEP reanalysis, except for air temperatures, which were provided by the Norwegian
Meteorological Institute at the Svea airstrip. The simulation ran for the period 1979 to 2018.

For a direct comparison between the model output and the observations, we extracted salinity and temperature
profiles from the simulation results for the same date on which an ice core was obtained during a field
campaign. With a fixed number of layers in the CICE model, layer thickness changes as a function of the ice
thickness. We resampled the salinity and temperature profiles in 5 cm vertical sections, with the zero-depth at
the ice surface, and the maximum depth at the ice bottom. After grouping the ice cores into different degree-
day classes, we computed the ice property statistics.

### 3.    Results
### 3.1.    Observed seasonal evolution of sea ice

Figure 2b shows the ice thickness for Utqiaġvik and Van Mijen Fjord during the growth season. With up to
4500 FDD, the ice growth season was longer and air temperature lower in Utqiaġvik than in Van Mijen Fjord,
with up to 2500 FDD and thinner ice in Van Mijen Fjord. Thus, in terms of ice melt, the main difference between
both locations (Figure 2c) was the ice thickness at the onset of melt. The maximum ice thickness was 1.20 ±
0.07 m in Van Mijen Fjord and 1.68 ± 0.11 m at Utqiaġvik.

Figure 3 illustrates typical salinity and temperature profiles for two particular DD intervals at Utqiaġvik: 3500−
4000 FDD for the growth season (a-d) and 10–15 TDD for the melt season (e-h). Following traditional
representation in sea ice, we initially examined ice core data with all profiles aligned at the ice surface
(Figure 3a-b, e-f). Overall, salinity profiles from different years showed larger variability in the lower third of the
ice cover (Figure 3a, e), with maximum salinity values above 8‰ at depth within 0.3 m of the ice bottom. By
aligning the profiles at the ice-ocean interface (Figure 3c-d, g-h), this variability is strongly reduced and we
clearly see the increase of the salinity within the lowest 0.1 m of the ice profile during the growth season (c).

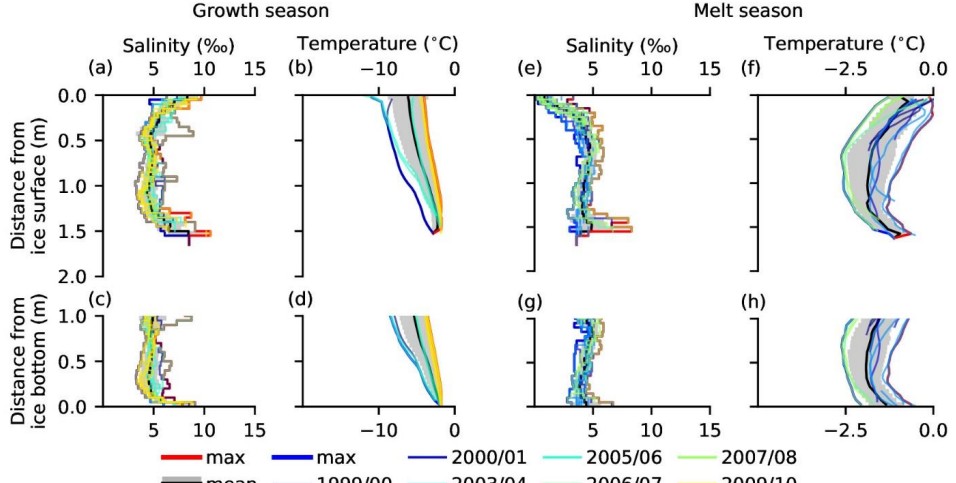

**Figure 3—Salinity and temperature profiles and statistical envelope for selected periods of the seasonal cycle in Utqiaġvik for (a) 3500–4000 FDD, typical range of the end of the growth season, with the typical C-shaped salinity profile; (b) 10–15 TDD typical of a few days after the onset of melt, with the typical inverted S-shaped salinity profile. Thin colored profiles correspond to different core observations, with colors indicating the year of the coring. The mean envelope, i.e. mean ±1 standard deviation, is shown as a thick black line with the gray area with the maximum and minimum value at each depth represented by red, respectively blue lines.**

Figure 4 displays the seasonal evolution of salinity and temperature for selected degree day (DD) intervals throughout a full seasonal cycle for Utqiaġvik. The full seasonal evolution including all DD intervals for Utqiaġvik and Van Mijen Fjord is available as supporting material for both observations and the CICE model simulations. Within the ice cover, variability was small, falling within a median standard deviation of 0.7‰ in Utqiaġvik and 0.9‰ in Van Mijen Fjord. Variability was larger in the upper 0.15 m and lower 0.2 m of the ice cover (1.2‰ and 1.1‰)

During the growth season, the salinity profiles exhibit the characteristic C-shape. The average salinity value of 5‰ in the internal columnar ice increases to higher values near the top and bottom surface (Figure 4a, 3500–4000 FDD). Temperature profiles are linear during the growth season (Figure 4b), constrained at the ice bottom by the freezing temperature of sea water, and controlled at the ice surface by surface heat exchange, which typically scales with air temperature and is further modulated by the presence of snow. In Utqiaġvik, we observed steep temperature gradients ($\Delta T/\Delta H_i > 10$ °C m$^{-1}$) during the initial phase of the ice growth (1500–2000 FDD). Towards the end of the growth season (3500–4000 FDD), the gradients decline as the ice temperature responds to surface warming. In Van Mijen Fjord, smaller temperature gradients ($\Delta T < 10$ °C m$^{-1}$) were observed throughout the growth season due to the milder climate.

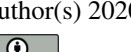

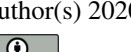

**Figure 4—Seasonal evolution of salinity (a, c) and temperature (b, d) for selected degree day intervals at Utqiaġvik (a, b) and in Van Mijen Fjord (c, d). See caption for Fig. 3 for additional guidance. The numbers at the bottom of each panel indicate the number of years of samples, and total number of samples (in parentheses).**

At the beginning of the melt season, desalination is observed in the upper layer (Figure 4a, 0–5 FDD), leading

to the inverted S-shaped profile typical of melting ice. Temperature increases in the upper layer are linked to





increases in ice porosity and onset of surface melt (Figure 4b, 0–5). With the temperature at the ice bottom
constrained by the seawater temperature, we observed a warming lag in the ice interior, leading to a C-shaped
temperature profile.

Later in the melt season (Figure 4a, 15–35 TDD), desalination progresses throughout the ice cover: average
salinity decreases to an average of 3.7 ± 0.6‰. There are minimum values close to 0 near the ice bottom in

the 25–35 TDD interval (Figure 4a). In parallel, meltwater flushing through the ice accelerates warming of the
ice cover. Finally, the temperature inverts with a near linear profile from 0 °C at the top to the seawater freezing
point at the bottom.

Due to the larger number of measurements at Utqiaġvik, DD intervals are comprised of at least 4 different
years of data. In contrast, only about half of the intervals for Van Mijen Fjord are 2 or more years (Figure 4).

In addition, the number of samples for each interval is significantly smaller for Van Mijen Fjord than for
Utqiaġvik, with the exception of the salinity composite profile in the 1000–1500 FDD interval.

### 3.2.    CICE Model Results

The envelope defined by the minimum and maximum simulated ice thickness for Utqiaġvik is plotted in
Figure 2b for the period 1998 to 2014. Ice growth observations at Utqiaġvik and in Van Mijen Fjord fall largely

into this interval. Therefore, the model seems to replicate the seasonal trend and variability of ice growth
correctly. During the melt season, the model tends to underestimate the ice thickness, as observations fall
closer to the maximum simulated ice thickness in Utqiaġvik. Figure 5a and 5b display modelled versus
observed ice thickness and snow depth, respectively. Overall, the model represents the observed ice thickness
fairly well with an average R = 0.73 for both locations, R = 0.77 for Utqiaġvik, and R = 0.67 for Van Mijen Fjord.

Considering only the growth season, correlation for ice thickness increases to R=0.89 for both Utqiaġvik and
Van Mijen Fjord. However, large discrepancies exist for the snow depth (R = 0.27 for both locations, R = 0.24
in Utqiaġvik, R = 0.31 in Van Mijen Fjord).

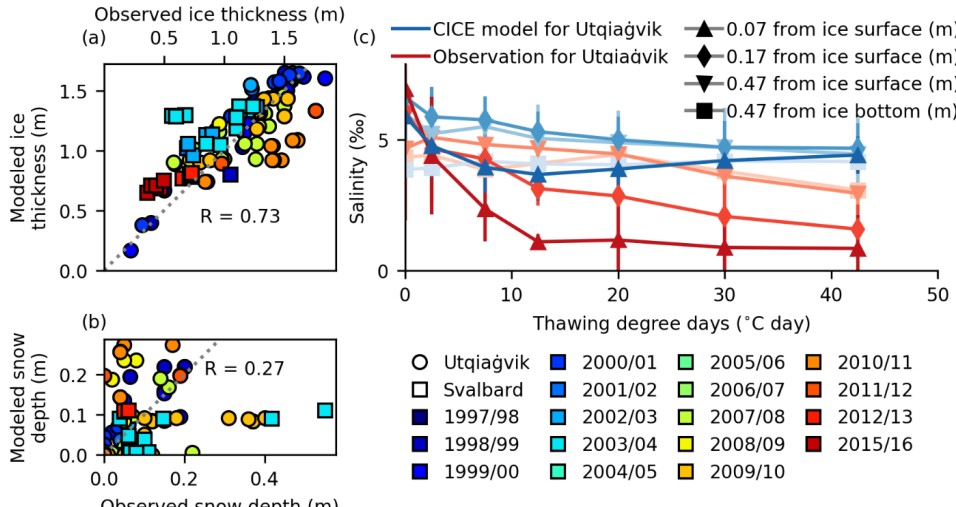

**Figure 5—Comparison between CICE model simulations and observations: (a) ice thickness, (b) snow depth, and (c) evolution of the salinity during the melt season at different depths; the salinity error bars indicate 2 standard deviation.**


Figure 6 provides the difference in mean salinity and temperature between the model and the observations for both locations. Table 1 summarizes the correlation coefficients between simulation and observation. We divided the ice cover into three layers: the upper layer, from the ice surface to 0.15 m depth, which corresponds approximately to a layer of granular ice; the lower layer, from the ice-water interface to 0.2 m upwards into the ice cover, corresponding to the actively growing layer with higher salinity; and the layer in between which we refer to as the interior ice cover.

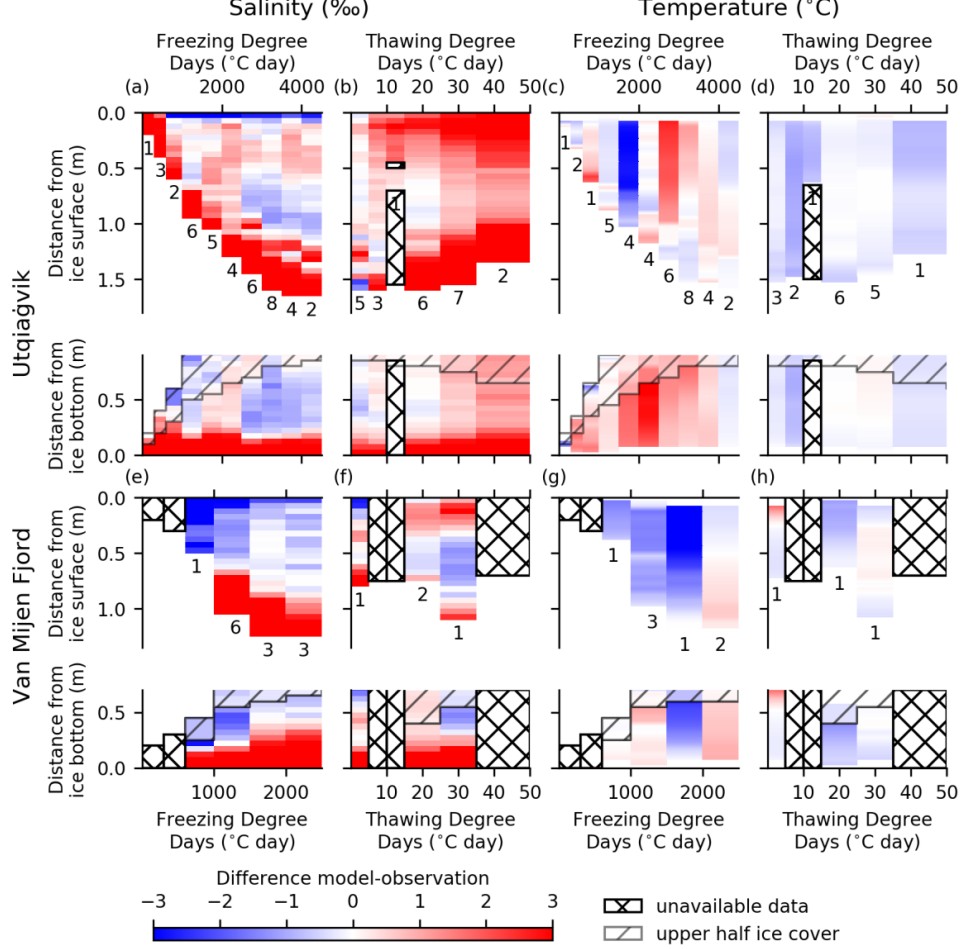

**Figure 6—CICE model performance for mean salinity (a-b, e-f) and mean temperature (c-d, g-h): difference between model output and observations for Utqiaġvik (a-d) and Van Mijen Fjord (g-h) during the growth (a, c, e, g) and ice melt (b, d, f, h) seasons. Cross-hashed areas show missing data, slash-hashed areas in bottom-aligned cores depict the upper half of the ice cover. The number below the column indicates the number of sampling years in the DD intervals.**

Overall, the simulated temperature field is strongly correlated with the observations (R>0.80), with the exception of the growth season in Van Mijen Fjord (0.25 <R <0.5). We expected some temperature fluctuation across the intervals (Figure 6c, e.g. 1500–2000/2000–2500 FDD), typically during the growth season, as the ice reacts readily to air temperature changes. The differences are larger at the ice surface and decrease away





from it, as the ocean temperature is almost steady at the ice bottom. During the melt season, the model tends to underestimate the temperature (Figure 6d, h). In Utqiaġvik, there are small positive temperature biases during the growth season (+0.1 °C) and negative during the melt (−0.1 °C). In Van Mijen Fjord, the model underestimates the temperature by approximately −1.4 °C, and presents a small positive bias during the melt (+0.1 °C). Standard deviations within the model and the observations are similar.

**Table 1—Pearson's correlation coefficient between model output and observation as a function of ice depth.**

| Layer | Utqiaġvik | | | | Van Mijen Fjord | | | |
| | Salinity | | Temperature | | Salinity | | Temperature | |
| | Growth | Melt | Growth | Melt | Growth | Melt | Growth | Melt |
|---|---|---|---|---|---|---|---|---|
| upper[1] | -0.17 | 0.14 | **0.97** | **0.74** | -0.59 | **0.68** | 0.56 | **0.97** |
| interior[2] | **0.76** | **0.23** | **0.96** | **0.91** | **0.72** | **0.66** | 0.31 | **0.64** |
| bottom[3] | **0.56** | **0.68** | **0.52** | **0.68** | 0.52 | **0.31** | 0.33 | 0.95 |
| Ice cover | **0.70** | | **0.98** | | **0.81** | | **0.80** | |

Coefficients shown in bold are significant at the 5% level.
[1]: upper 0.15 m, computed with profiles aligned at the ice surface
[2]: interior layer, computed with weighted average profile aligned at the top/bottom
[3]: bottom 0.2 m, computed with profiles aligned at the ice bottom

The salinity field during the growth season is overall well represented by the model (Figure 6a, c). Within the ice cover, the simulated salinities are strongly correlated with the observations both in Utqiaġvik (R =0.76) and Van Mijen Fjord (R =0.72). In the upper layer, the consistent underestimation of the salinity by the model is associated with a negative bias (Table 1). In the bottom layer, the model overestimates salinity by up to 10‰ during the growth season (Figure 6a, b, e, f, upper plots). When the profiles are aligned at the bottom, the vertical extent in which the model overestimates the salinity decreases from more than 0.4 m to less than 0.2 m (Figure 6a-b, e-f, lower subplots).

During the melt season, the model failed to capture the salinity field in both locations. The difference between the model and the observations increases as the melt season progresses (Figure 6b, f). The divergence is especially visible in the upper half of the ice cover. Figure 5c displays the evolution of the salinity during the melt period at three different depths in Utqiaġvik. In the uppermost layer (0.07 m and 0.17 m from the surface), we observed a rapid decrease of salinity from 7‰ to a plateau of 2‰ in fewer than 20 TDD. At the same depth, the model starts the desalination with the salinity decreasing slowly from the initial 6‰. However, soon after reaching 4‰ at 10–15 TDD, salinity increases again asymptotically to a finite value near 5‰. Lower in the ice cover, desalination occurs after 10–15 TDD (0.47 m from the ice surface). The salinity values decrease from 5‰ to 3.5‰ by 45–50 TDD. In the model, the salinity remains similar, and tends towards the same asymptotic value as in the upper layer.

In general, the modelled temperature field captures the seasonal variability (Figure 6c-d, g-h), with small anomalies (ΔT <1 °C) and good correlation (R>0.6), with the exception of the growth season in Van Mijen Fjord (R=0.31). During the melt season, the two model runs tend to underestimate the temperature.





Figure 7 shows interannual variability between observation and model from the winter 1998/99 to 2014/15 at Utqiaġvik. Overall freeze-up dates are well correlated between model and observations (R=0.68, p=0.048). An unusual breakout event happened on December 14th 2005. The landfast ice broke from shore and drifted away, leading to the initiation of new ice growth. For this particular ice, modelled ice thickness is overestimated relative to the observation.

Once the ice reached maximum thickness, model and observation agreed to within 10% (Figure 7b). Pearson's correlation for ice thickness, computed on the entire Utqiaġvik dataset, is R=0.87 (p <0.05). The model overestimated ice growth during the winter 2004/05 and 2005/06 by ~20% and underestimated the growth during the winter 2010/11 and 2011/12. Variability between the difference with measurements from the mass balance station (MBS, Figure 7b), and the ice core lengths remains small.

At maximum ice thickness, model snow depths do not fit the observations well (Figure 7c). At the end of the growth season, for 8 out of the 16 winters, modeled snow depths are either twice or half as large as the observations with large discrepancies between snow depth measured by the mass balance station and during the ice core collection. Overall, the modeled ice thickness is weakly correlated with the observation (R=0.32, p <0.05, Figure 5b).

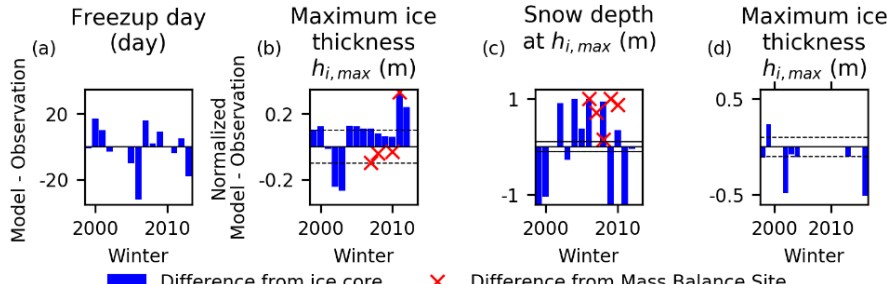

**Figure 7—Differences between model output and observations at Utqiaġvik (a, b, c) and in Van Mijen Fjord (d) for (a) freeze-up dates; (b, d) maximum ice thickness, normalized relative to observations; (c) snow depth at the maximum ice thickness, normalized relative to observations. Blue bars are based on (a) observation made by local experts or (b-c) collected ice cores. Red crosses show measurement at the mass balance site.**

Figure 8 depicts the relative error in computing the heat capacity for sea ice between modelled and observed temperature as a function of the ice temperature for 5‰ bulk salinity sea ice. We used the specific heat capacity as described by Ono's equation (Petrich & Eicken, 2017, Equation. 1.25). The sensitivity to temperature of sea ice increases with ice temperature. During the melt season, or when temperature is above -5 °C, a small difference in temperature of the order of 0.1 to 0.3 °C is enough for the modelled heat capacity to deviate by more than 20%. However, at lower temperatures (T < -10°C), a temperature difference between model and observation of up to 3 °C becomes negligible (relative error $\varepsilon_r$<0.1).





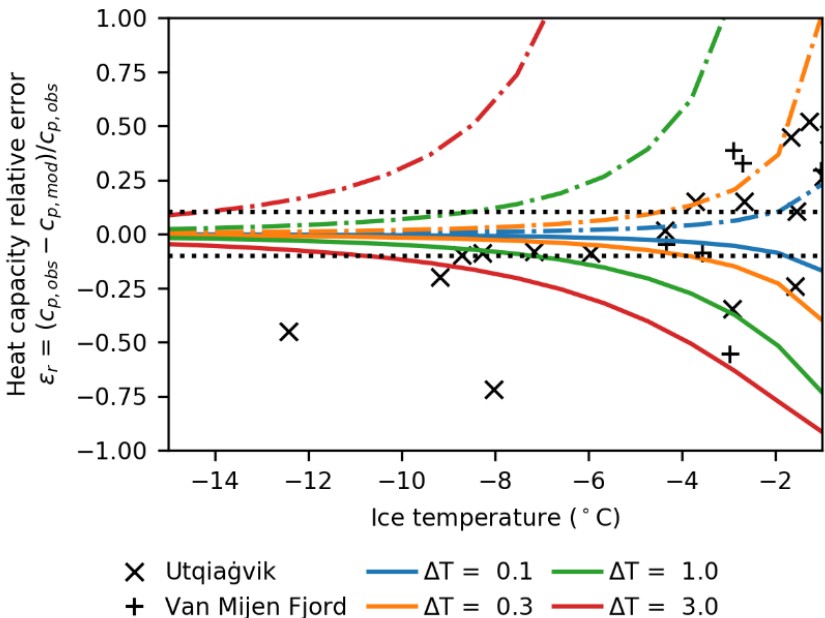

**Figure 8 – Relative error between heat capacity between observation and model for Barrow (x) and Svalbard (+). Colored lines indicate error for given temperature difference between observation and model of 0.1, 0.3, 1.0 or 3.0 °C and bulk salinity of 5‰. Bulk salinity for Barrow samples are 4.9‰±1.2‰ Sensitivity of heat capacity to smaller temperature difference increases with ice temperature.**

Figure 9 depicts the difference of the brine volume fraction between observation and model output in terms of natural variability and bias. During a cold growth season with an ice porosity of less than 3%, typical of Utqiaġvik (Figure 8a), the deviation of the brine volume fraction from natural variability remains small
(SD($V_{b,f}$)<0.3%). This holds true despite a large standard deviation for both salinity (SD(S) = ±0.9‰) and temperature (SD(T) = ±1.6 °C) with a small bias (ΔS = +0.4 ‰, ΔT = +0.1 °C). Before the onset of melt, in Utqiaġvik, or during a milder winter in Van Mijen Fjord (Figure 8a), ice porosity increases up to 5%. In Utqiaġvik, despite a decrease in the standard deviation in the model (SD(S) = ±0.7‰, SD(T) = ±1.0 °C), and an identical bias, their combination increases the porosity range by 1% for the model values relative to natural sea ice. In
Van Mijen Fjord, the range of porosity is similar between the simulation and the observation. However, the model bias shifted the porosities to lower values. During the melt season with ice porosity >12% (Figure 8c), we computed an identical standard deviation for both locations (SD(S) = ±0.7‰ and SD(T) = ±0.3 °C). However, temperature and salinity bias increased to ΔS = +1.2 ‰ and ΔT = −0.4 °C at Utqiaġvik, and ΔS = +0.5‰ and ΔT = -0.2 °C in Van Mijen Fjord. The range and values of porosity from model output are very
similar to the natural variability at both sites, though the bias shifted the intrinsic properties of the modelled ice cover.



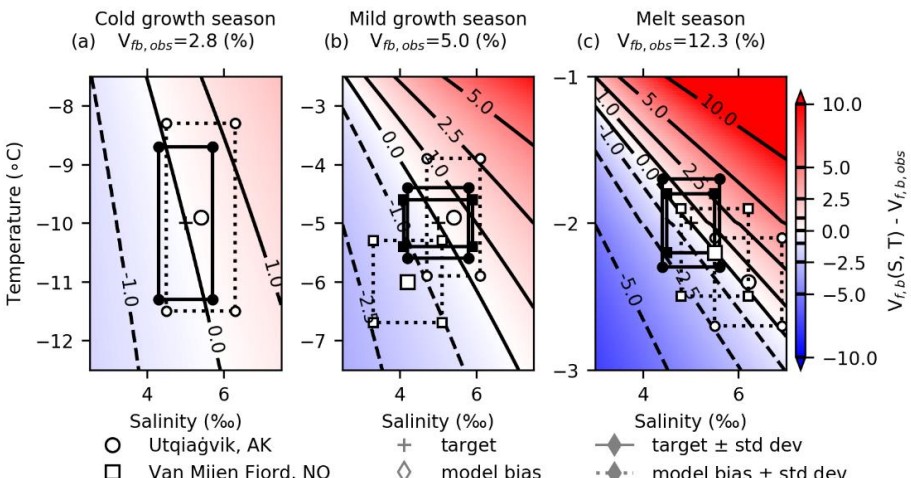

**Figure 9—Differences from observation (+) of the brine volume fraction as a function of model bias and standard deviation for (a) cold growth season with observed salinity S=5‰, and temperature T=-10 °C, (b) mild growth season with observed S=5‰, and T=-5 °C and (c) melt season with observed S=5‰, and T=-2 °C for Utqiaġvik (square) and Van Mijen Fjord (circle). The background displays the difference between brine volume fraction of an observation Vfb, obs and brine volume fraction Vfb (S, T) for a given salinity S and temperature T. The boxes depict the standard deviation from the observation (+, at the plot center). The dotted line represents the standard deviation from the modeled ice (white symbol).**

## 4. Discussion
### 4.1. Discussion of error and uncertainties
#### 4.1.1. Observations

The statistical analysis for Utqiaġvik was supported by a large dataset composed of 167 cores evenly spread throughout the season (Figure 4a, b). In comparison, we were only able to compile data from 60 cores for Van Mijen Fjord, with a majority falling into only three DD intervals (Figure 4c, d). A large number of profiles and sampling years divided by DD intervals not only minimizes the impact of any outlier profiles, but also makes it possible to quantify and explain the importance of an outlier in the time series. For example, at Utqiaġvik, the

early-winter breakout in 2005/06 led to thinner ice than in other winters. In contrast, the only core classified as the earliest stage melt season in Van Mijen Fjord (Figure 6f, h) is associated with a cold temperature profile, explaining the absence of desalination at the ice surface . Thus, the sea-ice climatology for Utqiaġvik is more complete and representative of the average local ice conditions throughout the whole seasonal cycle, than Van Mijen Fjord. In Van Mijen Fjord, the intervals 1000–1500 FDD, 1500–2000 FDD, 1500–2500 and 15–25

TDD contain enough cores to be representative. When comparing model output to observations, we only consider intervals with more than two sampling years.

At Utqiaġvik, cores were collected in two areas within 10 km of each other. Both sampling areas were in protected locations recognized for uniform ice growth (Druckenmiller et al., 2009), hence the datasets are very consistent throughout the year. In Van Mijen Fjord, six coring locations were spread over 40 km. Høyland

(2009) described two different growth regimes: In the lower part of the fjord, ice grows thinner due to a larger ocean heat flux; in addition, snow contributes to the formation of a combination of superimposed and snow ice



on top of the sea ice (Høyland, 2009). Accordingly, the variability in ice thickness within a DD interval is much larger in Van Mijen Fjord than in Utqiaġvik. However, the variabilities in salinity and temperature in Van Mijen Fjord are not significantly different from those at Utqiaġvik. Consequently, any additional data for Van Mijen

Fjord would significantly improve the representativeness of data with respect climatology.

Ice cores of different lengths were aggregated in the same group through the DD classification. Overall, standard deviation in the same group is less than ±10% of the average ice thickness. By aligning the ice cores at the surface, discrepancies accumulate in the lower third of the ice cover: during the winter 1999/00 and 2009/10, salinity of approximately 9‰ was found in the lower 5 cm sections for ice thicknesses of 1.55 m and

1.35 m, respectively (Figure 1a). To account for variations in ice thickness and allow direct comparison, Eicken (1992) proposed normalizing profiles with respect to the core length. The principal drawback of this method is that it disproportionally distorts layers in which large changes of salinity occur non-linearly over small vertical distances, typically at both interfaces.

Following the approach of sea-ice biologists and recent oil-under-ice experiments (Manes and Gradinger,

2009; Oggier et al., 2019), we aligned the ice cores at the ice-ocean interface, creating a well-defined L-shaped profile in the lower part of the ice (e.g. Figure 1c), but shifting the discrepancies discussed above to the top ice surface. By only considering the upper or lower half of the ice cover, depending on the alignment of the ice core at the top or bottom, and focusing the analysis accordingly, we provide a more accurate picture of sea ice processes. However, as this method required resampling the ice cores from the ice bottom upwards,

extreme values were smoothed during the process. In 2009, the maximum salinity value measured at the ice bottom was 11‰ in the lowest, 3 cm-thick section. After resampling into 5 cm sections from the bottom, the value decreased to 9‰ (Figure 1a, c). This issue would be easily solved by sectioning the ice cores in sections with a vertical 0-reference set at the ice surface for the upper half and at the ice bottom for the lower half, with an odd length section at the center of the ice cover.

Systematic errors associated with the coring method impact salinity measurements. Cox and Weeks (1975) describe brine drainage associated with core sampling, which depends on the porosity and permeability of the sample. During the growth season up to, and including, the onset of melt, brine loss is mostly limited to the bottommost 10 cm of ice, with average measured bulk salinity of 8 to 10‰ (Figure 4a, 0–4000 FDD and 0–5 TDD). Both salinity values and brine loss depth correspond to previous observations (Cox and Weeks, 1974;

Griewank and Notz, 2013, 2015; Notz et al., 2005). Using an *in-situ* impedance instrument, Notz & Worster (2009) demonstrated that a salinity profiles were continuous through the ice-ocean interface and the bulk salinity increased from about 5‰ 10 cm away from the ice-ocean interface to 32 ‰ 2 cm away from the interface, with seawater salinity 34‰. The combination of local temperature and bulk salinity near to those of the ocean underneath result in a high brine volume fraction at the growth front, which in turn translates to a

very porous microstructure. Thus, as the core is collected, a non-negligible amount of brine flows out of the core, and leads to a systematic decrease of the bulk salinity of ~20 ‰ in the lower 5 cm, and ~10 ‰ in the layer above. As this error is related to the sampling method, modelled salinity profiles do not suffer from this error, and exhibit higher salinity at the ice bottom (Supplemental Material Section S1-b, S2-b, S4-b). After the onset of melt Figure 4a, 5–25 TDD), the start of bottom melt and subsequent freshwater flushing explains the



decrease of average salinity from 8‰ to less than 5‰ at the bottom interface (Eicken et al., 2004; Notz and Worster, 2009).

With more than 30 researchers involved in the collection of ice cores in Utqiaġvik, differences between individual core handling and sample processing approaches may have caused different levels of brine loss. Thus, accuracy in measuring the bulk salinity in samples with high porosity is likely to vary more from one team

to another than in samples with low porosity. The standard deviation in our data increases from 1‰ within the ice cover to up to 4‰ in the bottommost layer. However, the 69 ice cores collected and analyzed by Gough et al. (2012) display small variability (≈1‰), including in the lowermost layer (Gough et al., 2012, e.g. Figure 9a). We estimate that at least 2‰ of the higher variability at the ice bottom may be to differences in sample handling. Uncertainties and errors in temperature measurements are more difficult to quantify. Near the freezing point,

small changes in temperature have a stronger impact on ice porosity and microstructure. Accordingly, small variations in temperature measurements may lead to large differences in the associated physical properties. The incorporation of temperature profiles, known to be measured from the ice bottom upwards, and not from the coldest to the warmest, has a negligible effect during the winter, but may have led to a positive shift of up to 0.2 °C at the onset of melt.

Single temperature outliers in a profile are mostly due to fractures in the ice core, which can accelerate local temperature changes. The bias value could be either corrected using a weighted average between the two neighboring measurements or a median filter on the grouped profiles. Other ice cores with non-linear behavior or a temperature outlier not associated with a fracture were compared with profiles measured by thermistor strings to verify the profile integrity. In rare cases, after careful review, some profiles were not incorporated in

the collection, as we suspected measurement errors.

During the melt season, quantifying uncertainties and errors is particularly challenging. Profiles with some temperature measurements above 0 °C are good indicators of the lack of effective solar radiation shielding during measurements; a heated probe can raise the measured temperature by several degrees. However, temperatures between the freezing point of sea water (−1.86 °C) and 0 °C are physically plausible.

Temperature variations are small across profiles from the same year (Figure 6, e.g. 2000/01, ΔT=0.2 °C), though large differences can propagate throughout the ice cover in between years (Figure 6, e.g. 2000/01 vs. 2006/07 ΔT = 0.3 °C). The convergence of temperature profiles to a single seawater temperature during the growth season indicates that there was no systematic calibration error in the probe throughout the year (Figure 6b,d, Figure 7b). In addition, the C-shaped temperature profile could only be achieved by warming

both ice interfaces. Thus, the large standard deviation is representative of interannual variability, and uncertainties are comparatively small.

Lack of temperature measurements during the melt season in Van Mijen Fjord does not allow for any uncertainty discussion for this location, and we assume the respective issues are comparable to Utqiaġvik.

While landfast ice will never be the predominant type of ice in the Arctic, the combination of an increasing

amount of open water (Barnhart et al., 2015) and decrease of perennial ice (SOURCE) will likely yield a higher ratio of first year ice versus multiyear ice. In addition, deformed ice accounts for 50-70% by mass according to Flato & Hibler (1995), which translates to level or undeformed ice accounting for at least one third, if not more





than half of the sea-ice extent. As such, the findings of our study are and will remain valid for a large part of the Arctic.

### 4.1.2. **Model**

In evaluating CICE model performance, small deviations in both temperature and salinity become more important as the ice approaches freezing point of seawater during spring warming. In order to discuss model performance, we need to determine the range of acceptable differences between observations and simulations with respect to salinity and temperature, and their evolution. Warming and phase transition of a material are

best related to its specific heat capacity, which is defined as the energy required to heat the material by one degree centigrade. We assumed that a 10% difference between the heat capacity computed from observations and the model is acceptable. According to Figure 8, for a 5‰ bulk salinity, the acceptable difference in temperature decreases exponentially from about 3 °C at an ice temperature of -15 °C to 0.3 °C at -5 °C, and falls below 0.1 °C above -2 °C; for ice at a temperature -5 °C and a bulk salinity of 5 ‰, the acceptable salinity

difference is about 0.7‰. Most of the uncertainties in the model output are related to the input data. Although uncertainties are difficult to quantify, the simulations for Utqiaġvik are assumed to be more accurate, as most of the forcing data are available locally. In Van Mijen Fjord, only historic temperature data are available at Sveagruva; all other data are supplemented from reanalysis. With a grid size of 2.5° x 2.5 °, the area covered by the reanalysis data is 3 times as large as Van Mijen Fjord, and about 50 times larger than the area of

interest in Utqiaġvik, and does not capture local variations, especially in rough terrain such as fjords.

We calibrated the model parameters to best replicate the observed ice growth. In the absence of historical records, some parameters were defined as constant for each season, although they can vary on a daily basis. Overall, the variations are not large enough to strongly impact the seasonal evolution. However, in the most recent years, significant bottom-melt occurred in Utqiaġvik before the end of the growth season (Eicken, 2016).

An increased ocean-to-ice heat flux is the most likely explanation for those early melt events and our current parameterization is unable to reproduce such events, leading to an overestimation of ice thickness.

### 4.2. **Sea-Ice climatology**

We were able to assemble a sea-ice climatology for salinity and temperature. The aggregation by degree days, rather than ice thickness or time, allows for a fully automated classification as a function of the seasonal

development stages of the ice: growth and melt, with sub-categorization by length (growth) or age (melt).

As expected, ice growth can be approximated with a quadratic growth function of FDD, with R=0.83 independent of location (Figure 2b, Leppäranta, 1993). Following Petrich et al. (2012), specific parameters for each location improve the goodness of the fit to R=0.89. Main differences in ice growth are related to the timing of the freeze-up in each location. Although freeze-up occurs at different times of the year, in December in Van

Mijen Fjord, and from late October to mid-December in Utqiaġvik, the average air temperature during freeze-up is similar for both locations at −16.5 °C and −17.3 °C, respectively. However, net solar radiation ranges from −50 to 0 W m$^{-2}$, in Ny-Ålesund, Svalbard (Ørbæk et al., 2012; Yamanouchi and Ørbaek, 1995), and from 0 to 50 W m$^{-2}$ in Utqiaġvik (Dissing and Wendler, 1998; Maykut and Church, 1973). The larger net radiative loss in Van Mijen Fjord initially fosters a higher growth rate (Figure 2b). Thickening of the ice and snow

accumulation quickly decreases the growth rate.





The growth season in Van Mijen Fjord is not only delayed by the milder climate, but also ends earlier by about 2 weeks, resulting in only half of the FDD accumulated in Utqiaġvik. The combination of a shorter growth season and a higher ocean-to-ice heat flux from warm incoming Atlantic water (Mauritzen et al., 2013; Schauer et al., 2004) significantly limits the ice growth to thinner ice in Van Mijen Fjord, relative to Utqiaġvik. In addition
to the difference in maximum ice thickness of 0.5 m, fewer observations in Van Mijen Fjord, particularly from the growth season, limit the potential for comparison with Utqiaġvik.

### 4.2.1. **Utqiaġvik**

Seasonal evolution of the composite salinity profiles resembles the seminal figure of Malmgren (1927) (Figure 4a, b). During the growth season, typical C-shape profiles are observed at Utqiaġvik. Within the internal
ice cover, the slow but steady decrease of the salinity from 5.9 ± 1.6‰ at 300/600 FDD to 4.8 ± 0.5 at 4000/4500 FDD is attributed to salt loss by gravity drainage through brine channels that penetrate the low-permeability ice interior to the seawater below (see Cox and Weeks (1974) and Cottier et al. (1999)). In the lower layers with higher salinity, and hence greater local permeability, convective overturning is initiated when the Rayleigh number becomes supercritical (Wettlaufer et al., 1997) and contributes to the lowering of the bulk
salinity (Notz and Worster, 2009).

The increase of salinity in the upper layer from 5.8 ± 0.8 ‰ at 300/600 FDD to 8.9 ± 1.5‰ at 2000/2500 FDD may be the result of upward migration of brine through cracks and connected pores, as proposed by Ono and Kasai (1985) and Eicken (1992). However, the combination of thin ice (0.3 – 0.4 m) and average high bulk salinity of the ice cover (6 ‰) at 300/600 FDD contributes to a higher brine volume fraction, which in turn favors
brine loss through the whole vertical extend of the samples. Thus, the bulk salinity, including at the surface, of ice formed earlier in the season is likely to be biased towards lower values.

In our analysis, the interannual variability is well represented by the standard deviation. We used the median, rather than the mean, of the standard deviation within the internal ice cover to minimize the influence of any outliers. For the salinity, the median standard deviation is 0.7‰, the same order of magnitude found in previous
studies focusing on single-season observations (Gough et al., 2012; Tucker et al., 1984; Weeks and Lee, 1962), which suggests low interannual variability.

The fan-shaped composite temperature profiles indicate prevalence of a linear temperature profile between surface temperature and fixed seawater freezing point temperature at the ice bottom. Colder ice, containing a smaller fraction of liquid brine, has a lower effective heat capacity and is more directly coupled to atmospheric
conditions. Hence, standard deviations are larger near the ice surface and decrease with depth. Temperature at the ice surface can vary on the order of several degrees centigrade over a few days (Figure 6b, winter 2005/06, std = 1.2 °C, in the middle of the ice cover), which is of the same order as the interannual variability (Figure 4b, median std = 1.3 °C, in the middle of the ice cover). Thus, despite steady warming of the winters in Utqiaġvik leading to a measured average ice thickness decrease by 2 cm per year ($R^2$=0.31) from
2000–2016, potential impacts on the composite temperature, such as a positive shift of the mean temperature, or a larger standard deviation, are obscured by the signal of the interannual and seasonal variability. Continuous long-term monitoring at key locations, such as Utqiaġvik, is necessary to quantify the effect of climate change in the Arctic on sea-ice properties and their interdecadal and interannual variations.



The strong increase in the monthly average of net solar radiation from 50 to 150 Wm$^{-2}$ from March to April
(Dissing and Wendler, 1998) marks the beginning of the melt season, as it initiates the warming of the ice
cover (Figure 4b, e.g. 4000–4500 FDD).

At the onset of melt, solar heating in the upper layer allows the brine to melt an increasingly larger volume
fraction of the surrounding ice. While the phase transition absorbs the energy, no heat is transferred deeper
within the ice cover, resulting in a loss of the linearity in the temperature profile (Figure 4c, e.g. 0–5). Increasing
ice temperature is associated with an increase of the porosity, which in turn leads to an increase of the
permeability, up by 2 orders of magnitude (Eicken et al., 2002). Initially, meltwater resulting from snow melt
spread laterally in the upper layer of ice. As the permeability the increases within the internal ice cover,
freshwater percolates downward. Thus, the salinity profile evolving from the C to the inverted S-shape is the
result of the initial desalination in the upper.

During the melt period, while temperature increases (Figure 4c, e.g. 5–10), salinity decreases within the ice
cover to 3.7 ± 0.6‰. The combination of freshwater flushing throughout the ice cover and bottom melt
contributes to the loss of the high-salinity layer at the bottom, leading to an inverted S-shaped salinity profile,
similar to the ?-shape salinity profile described by Eicken (1992). Minimum salinity values close to 0‰ in the
lower layer (Figure 3c, 25–35 TDD) indicate freshwater underplating, which occurs when low-density
freshwater collects at the ice bottom, above the denser seawater. Although this phenomenon is well described
(Eicken et al., 2002), the rare observations at Utqiaġvik are not sufficient to visibly impact the composite
profiles.

The temperature variability of 0.3 °C, given by the median standard deviation, is similar to estimated
temperature uncertainties, ranging from 0.2 to 0.4 °C. Compared to the temperature range of melting sea-ice,
roughly from −5 to 0 °C, those values are not negligible. It is of particular importance to evaluate the impacts
of temperature variability and uncertainties on sea-ice properties due to the increasing sensitivity to
temperature as ice warms (Cox & Weeks, 1986; Golden et al., 2007).

### 4.2.2. Van Mijen Fjord
Despite the climatology lacking some DD intervals, seasonal evolution in Van Mijen Fjord closely resembles
Utqiaġvik, with respect to a shorter ice growth season.

During the growth season, the salinity within the center of the ice cover decreases from 6.7 ± 0.9‰ at 300–
600 FDD to 4.6 ± 0.8‰ at 2000–2500 FDD, with higher salinity at both interfaces delineating a C-shape profile
(Figure 4c). The higher salinities observed in Van Mijen Fjord are not significantly different from Utqiaġvik
(Student's t-test, p=0.75). The faster growth rate observed in Van Mijen Fjord at the beginning of the growth
season (Figure 2b) may result in saltier sea ice. Later the combination of a warmer growth season and a higher
ocean-to-ice heat flux, from the Gulf Stream and Atlantic water inflow, coupled with a thicker snow cover leads
to higher ice temperatures than in Utqiaġvik (Rudels et al., 2005; Schauer et al., 2004). The milder and wetter
climate of Van Mijen Fjord allows for two processes to modify ice salinity. On the one hand, heavier snowfalls
locally depress the ice cover, leading to surface flooding and the formation of snow ice, a high salinity layer.
On the other hand, rain events or refreezing meltwater on the ice surface can create a low-salinity layer of
superimposed ice (Figure 4c, 1000-1500FDD, ice surface). While both processes have been observed at





Utqiaġvik, they are more common in Svalbard (Gerland and Hall, 2006; Høyland, 2009). The negative trend in the salinity of the upper and central layers during the growth season suggests that warm spells favor the formation of superimposed ice rather than snow ice. However, our dataset may be too limited to draw definitive

conclusions. An assessment of the effects of different types of precipitation throughout the season would be necessary to determine which process dominates the change in salinity and quantify the occurrence. Such analysis is beyond the scope of this study.

Desalination after the onset of melt is particularly prominent in the upper layers. Thicker snow cover and the presence of superimposed ice in Van Mijen Fjord led to an increased amount of freshwater available for

flushing. Rain events during melt will further lower the salinity in the uppermost layer (Figure 4c, 15-35FDD, 0-0.15 m depth).

### 4.3. Comparison of Field Data and Model Results

Overall the model is better at recreating patterns similar to discrete observations for Utqiaġvik than Van Mijen Fjord. While the size of the dataset used to tune each model may explain a better parameterization of key

processes, we suggest that the milder, more variable climate of Van Mijen Fjord is more difficult to replicate in CICE's standalone mode. Utqiaġvik, located on the shore of the Arctic Ocean, lies far away from the ice edge during most of the growth season, and is far removed from weather systems in the North Pacific that could bring moisture and heavy snowfall. The Svalbard Archipelago on the other hand remains close to the ice edge most of the growth season, and the local weather is directly influenced by storms evolving in the North Atlantic.

The isolated location of Utqiaġvik makes for very similar and reproducible cold growth conditions from one year to another, which yields a very consistent climatology. In contrast, Van Mijen Fjord is subject to more complex weather patterns and quickly changing growth conditions, which may not be fully captured with a model run on a single cell.

#### 4.3.1. Interannual variability

The model captures the general trend of the interannual variability, with the exception of the snow cover (Figure 2b-c and Figure 7). Spatial and temporal heterogeneity of the snow makes both observation and modelling challenging (Webster et al., 2018). We observed large variations between snow depth measured at the mass balance site and snow pit data gathered preceding the core collection (Figure 7). We expected the strong correlation for the ice thickness, and the observations were used to tune the model parameterization.

In addition, ice thickness standard deviations are mostly smaller than 10% of the observed ice thickness, which is within the same order of magnitude as the bottom roughness of level ice (Weeks, 2010; Wilkinson and Wadhams, 2016).

At the beginning of the growth season, differences in the freeze-up date of up to 15 days, with a typical average daily temperature of −20 °C, lead to an increase or decrease of about 250 FDD. We found the impact of such

differences to be negligible when aggregating ice cores in 500 FDD intervals. Larger differences, as observed during the winter 2005/06, have a significant effect on the length of the growth season, with the addition of more than 455 FDD in 32 days. The unique grid cell on which the sea-ice seasonal cycle is simulated is isolated from any external influences that impact the seasonal cycle, such as ice movement, drifting snow or warm water upwelling. In particular for land-fast ice, the model is not able to capture breakout events or late freezing,





like during the winter 2005/06. This highlights the need for local freeze-up observations, in order to validate
the model output and correctly aggregate the ice core data.

### 4.3.2. Seasonal variability

Overall, seasonal variability is well captured by the model. The temperature field remains highly correlated
(R=0.9 in growth and melt season). During the growth season (Figure 6c, g), the sign changes of the
temperature anomaly as a function of FDD intervals is related to the weather at the time of the coring, rather
than representing an inconsistency in the model. After the onset of melt, the temperature is consistently
underestimated in Utqiaġvik (−0.45 °C) and in Van Mijen Fjord (−0.2 °C). The negative bias may be the result
of inaccuracies in the calculation of the down-welling long-wave radiation and shortwave fluxes, or due to solar
heating of the probe. Although profiles with clear evidence of solar heating were discarded, it is likely that both
factors contribute to the underestimation of the temperature field in the model.

During the growth season, the model captures the C-shape of the salinity profiles, as well as the average
salinity. The weaker correlation in the salinity field (R=0.7) is mostly driven by the difference in bulk salinity at
both interfaces (Figure 6a, e). In the lower layer, brine loss associated with the sampling artificially decreases
the bulk salinity of the observation. Thus, the modelled salinities are likely to be more representative of the
bulk salinity occurring naturally in the lowermost layer of the sea ice. The development of in-situ salinity
measurement methods (Notz et al., 2005) or improvements of the current sampling method are necessary to
ultimately assess the accuracy of the change in bulk salinity in the lower layer. In contrast, the salinity
underestimation in the upper layer suggests that the model does not capture upwards brine migration, which
causes increasing salinity near to the ice surface (Eicken and Lange, 1989; Niedrauer and Martin, 1979).

The modelled salinity field during the melt season is weakly correlated with observations (R=0.3). The model
failed to fully reproduce desalination, which is governed by multiple complex physico-chemical processes.
Upon the warming of the ice, brine inclusions grow and become interconnected, allowing both meltwater
flushing in the upper layer (Untersteiner, 1968) and convective brine overturning in the lower layer (Eicken et
al., 2002). In addition, the movement of brine and meltwater within the ice cover is associated with changes in
bulk salinity, and heat transport, which in turn modify ice microstructure and associated transport properties.
Despite simple descriptions of desalination deemed elusive in the context of earth-system models (Hunke et
al., 2011), progress has been made to incorporate flooding and flushing into parameterization schemes for
large-scale models (Griewank and Notz, 2015).

As sea-ice state variables, temperature and salinity are used in semi-empirical equations describing further
physical properties of sea ice (Cox & Weeks, 1983; Petrich & Eicken, 2017), which play an important role for
transport processes (Pringle et al., 2009) and biogeochemical exchanges in the Arctic ecosystem (Miller et al.,
2015). In the following we briefly discuss the impact of model bias and standard deviation on the brine volume
fraction—a common proxy for permeability and pore connectivity—to quantify potential impacts of model
uncertainties and natural variability on the computation of sea-ice properties at high ice temperatures, where
temperature sensitivity increases (Golden et al. (2007)).

The standard deviation of the model and the observations are similar, both in terms of magnitude and seasonal
evolution. With the onset of melt, as ice desalinates and warms up, the range of salinity and temperature





narrows, and the variability decreases. Nonetheless, due to the increased temperature sensitivity near the
melting point, the remaining combined variability in temperature and salinity lead to a significant increase in

the possible range of brine volume fraction and porosity. In Utqiaġvik, the standard deviation of temperature
during the growth season is 4 times larger than during melt. In contrast, the range of porosity is 4 times larger
during melt than in the ice growth season (Figure 8). In Van Mijen Fjord, the standard deviation of temperature
decreases by a factor of 3 from growth to melt, while the porosity spread increases from 2.5% to 7%. The
modelled salinity and temperature, dependent on the location and varying from season to season, may lead

to a significant difference in the brine volume fraction from the natural variability, such as in Van Mijen Fjord
during the growth season (Figure 9a, -1.5%). In other cases, particularly during the melt season, the difference
is almost negligible (Figure 9c). Model bias is independent of season and location and this impact is more
difficult to quantify. Our analysis suggests that a model bias remaining below 10% of temperature and salinity
results in acceptable values of brine volume fraction, with regard to standard deviation and natural variability.

**5.    Conclusions**
The analysis of more than 160 cores from Utqiaġvik allows us to derive a salinity and temperature climatology
data for first-year sea-ice representative of a cold and long growth season. Computing composite profiles from
salinity and temperature profiles aggregated by freezing and thawing degree days yield a statistical
representation of the natural variability in the different seasonal development stages. Our effort to build a

similar dataset for locations defined by a shorter and milder growth season was limited by the smaller number
of cores available from Van Mijen Fjord (60 total). We found that a composite profile should draw upon a
minimum of 3 sampling years, each with at least 2 salinity and 1 temperature profile, to ensure that it is
representative of average ice conditions.

Cores of different lengths introduced artificial discrepancies when performing statistical analysis, which

increases with the distance from the zero-reference horizon. Our method of dividing the ice cover in half, and
aligning the ice cores to the corresponding interface limited such problems. However, some information is lost
as we had to discretize and average sections of different heights. We suggest an alternate field sampling
method to be used in the future to improve the ice core data quality:  the upper half of the core is sectioned
with the ice surface serving as the reference horizon, while the lower half is sectioned with the ice bottom

serving as reference horizon. This greatly facilitates direct comparisons between cores. In addition, this
approach supports observations of both the ice surface, which is of particular importance for remote sensing
applications, and the ice bottom, important from a biological perspective and for fluid exchange considerations.
The impact of the resulting odd length section at the centre of the core is negligible, as bulk properties in the
ice interior typically vary much less with depth.

At both locations, salinity profiles conform to the typical ice salinity evolution patterns (C and S). We show that
no significant difference in salinity composite profiles exists between Van Mijen Fjord and Utqiaġvik with
respect to the development stage. From the growth to the melt season, salinity in the centre layer decreases
from 5.5 ± 0.8‰ to 3.7 ± 0.6‰, similar to commonly accepted salinity means and standard deviations. While
mean temperatures during the growth season are significantly colder in Utqiaġvik (−10.1 ± 1.3 °C) than in Van

Mijen Fjord (−5.0 ± 1.4 °C), temperatures during the melt season are similar (−2.0 ± 0.3 °C at 15–35 FDD).



We compared observed climatologies for each location with the prognostic salinity and temperature of the CICE model forced with specific local parameters. Overall, the model accurately replicates the sea-ice seasonal evolution and the natural variability of the ice thickness as well as temperature, and salinity during the growth season. However, based on the analysis shown in Figure 8, the CICE model failed to capture the

progression of sea-ice desalination during the melt period. The resulting positive salinity bias leads to an overestimation of the effective specific heat capacity, which in return alters the calculated surface heat fluxes, and impacts sea-ice porosity and transport property estimates.

Finally, we show that both the standard deviation and the effect of the model bias depend on location and vary from season to season. The combined impact of natural variability and model bias increases exponentially as

the ice approaches the freezing point of seawater. While current models successfully simulate much of the growth season, significant knowledge gaps remain concerning the driving processes of the melt season. Simulation and prediction of the seasonal evolution of sea-ice are of increasing importance to constrain how changes may impact Arctic climate and ecosystems. Systematic and continuous monitoring of sea-ice conditions in different climatic zones is needed to improve model performance and process understanding.





## 6. Author contribution

M. Oggier and H. Eicken designed the study and helped lead collection of field data in Alaska. K. Hoyland led data collection in Svalbard. M. Oggier processed observational data and model output, performed the analysis, drafted the manuscript and designed the figures. M. Jin collected the forcing data for the model and ran the CICE Los Alamos model. H. Eicken and K. Hoyland aided in interpreting the results. All authors discussed the results and contributed to the manuscript.

## 7. Competing interests

The authors declare that they have no conflict of interest.

## 8. Acknowledgements

The authors acknowledge the National Science Foundation for funding support through the CMG Program (OPP-0934683), supplemented by support of the SIZONet Project (OPP-0856867) for work in Utqiaġvik. In addition, data collection in 2016 in Van Mijen Fjord and the written works were partially supported by the Research Council of Norway PETROMAKS2 program (MOSIDEO—Project Number 243812). The authors wish to acknowledge the support to the FATICE project from the MarTERA partners, the Research Council of Norway (RCN), German Federal Ministry of Economic Affairs and Energy (BMWi), the European Union through European Union's Horizon 2020 research and innovation programme under grant agreement No 728053-MarTERA and the support of the FATICE partners. Data analysis was supported by a Graduate Research Fellowship funded by the Oil Spill Recovery Institute, Cordova, Alaska. Thanks to UIC Science/Umiaq for field logistics in Utqiaġvik. We would like to thank the current and past members and visitors of the Sea Ice Group at the University of Alaska Fairbanks, among others Andy Mahoney, Josh Jones, Chris Petrich, Megan O'Sadnick, Mette Kaufmann, as well as collaborators Kyle Dilliplaine, Mats Granskog, and Chris Polashenski. Thanks to Store Norske Spitsbergen Kullkompani for providing logistical support in Svalbard, and to all the scientists and students involved in the data collection in Van Mijen Fjord, among others Martina L. Solomon, Åse Ervik, Per Olav Moslet. Lea Hartl and Allison Fong have improved the readability of this paper, their help is much appreciated.

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
