# Peer review of "Seasonal and interannual variability of sea-ice state variables: Observations and predictions for landfast ice in northern Alaska and Svalbard"

_The Cryosphere, 2020_

## Referee Comment (RC1) · Anonymous Referee #1 · 9 Apr 2020

In the submitted manuscript Oggier et al have analyzed 180 fast-ice cores from Alaska and 60 ice cores from Svalbard gathered over roughly a decade. The cores are binned together by degree days (a unit the authors use instead of time to sort the cores into differing stages of the sea-ice life cycle), and various properties of the ice are discussed in regards to the sea-ice's life cycle and how much they vary from year to year. At both study locations simulations are run using the 1D CICE sea ice model, and the model output is compared to the ice core data and other measurements taken from the many measurement excursions over the years.

Given that the paper discusses sea ice in detail, it definitely falls within the scope of

[Figure]

TC. The novelty of the paper lies less in the data and simulations used, and more in the methods used to compare sea ice from differing times and of different thickness. The many cores in addition to the model simulations provide the authors with a wealth of data to draw conclusions from. However, I find that the authors struggle to distil new insights from this wealth of data. A lack of clear scientific questions made it difficult to judge if the methods used are suitable, and neither the introduction nor the structure of the paper give the reader a sufficient frame of reference to follow. I am unable to distinguish when the authors summarize what has already previously been known from when the authors are introducing their own results.

In addition to the missing storylines and poor flow of the paper, the figures of the manuscript are extremely busy and difficult to process. The colors chosen are difficult to distinguish and not colorblind friendly, and data is often obscured by overlapping lines/dots. A further issue is that the authors do not follow the TC data policy. I found no statements regarding the availability of the data used, nor a link or reference to the precise model version of CICE used to run the simulations.

For the reasons listed, I recommend that the paper be rejected. However, since the data itself is solid and because there are many interesting facts scattered throughout the submitted manuscript, I strongly encourage the authors to refine the aims and scope of the manuscript and then resubmit. My impression of the submitted manuscript is that it attempts to cover too many things at once.

The remainder of the review will raise some general issues I found particularly problematic, followed by detailed comments on the individual figures.

General issues

Missing questions

Currently, the paper introduction raises no questions. It simply states that data is needed, and that the authors provide data. If this is the case this paper should be reformulated as a technical description or data paper. There are a wealth of questions that could be raised. Here just two examples:

- The decade long collection of ice cores in Alaska is unique in the number of cores gathered and the time covered. However, it is unclear if the data contains additional variability in addition to the interseasonal and spatial variability due to the constantly changing participants who extracted the cores. In this paper we .....

- Reference profiles of salinity are commonly provided in normalized coordinates from ice-snow interface to ice-ocean interface (e.g. lots of citations). This approach functions poorly for first year ice which changes thickness rapidly. In this paper we will determine if providing reference profiles in meters from the ice-ocean interface is more suitable for studies of ice biogeochemistry.

Model-observation comparison

The authors failed to convey what is gained by including the model in this paper. The inclusion of the model is further complicated by the authors not cleanly separating what is used to force/tune the model versus what is used to evaluate it. The ocean heat flux was tuned to fit the ice depth, and then the ice thickness was used to evaluate the model performance? It has been known since the 60s that ice thickness is dominated by ocean heat flux, atmospheric heat flux, and snow depth. Accordingly evaluating

simulated ice depth says less about the model than the forcing data. Are the authors attempting to verify the consistency of the forcing data? All aspects of the model are studied in much greater detail in other papers, for example Lecomte et al 2013 in regards to snow. Are they attempting to evaluate the salinity parametrizations? If so they should refer to and frame their results in regards to recent research in that area, e.g. Max Thomas et al 2020 "Tracer Measurements in ....", or Jacob Buffo et al 2018 "Multiphase reactive transport and platelet Ice Accretion ..."

Similarly, why are the authors looking at ice heat capacity during melting? It is already known from basic sea-ice thermodynamics that the heat capacity is very sensitive to changes in salinity and temperature close to the melting point, no model or observations are needed to confirm this. The heat capacity also has very little impact on simulating ice melt compared to the completely dominating effect of the surface albedo. I personally found the modelling aspect of the paper very unconvincing, and would encourage the authors to figure out exactly how the model helps them convey their results.

Structure

I would highly recommend that the authors rethink their current approach of having one big results section, followed by a very long discussion section. It is also not helpful that the current results section is predominantly filled with descriptions of figures. By just describing data in "results" without a purpose the reader has no guidance what is important. And then when the authors raise points in the discussion many pages later the reader has already long forgotten what they saw. I recommend going through the scientific questions one by one, and supply the specific data needed to address each specific question as it is being discussed.

Climatology

Despite how often it is referred to I do not know what the authors mean by climatology. It initially sounds like they are attempting to provide a reference set of profiles for others to use, similar to a sea surface temperature climatology map. But by the time we have reached section 4.2., "climatology" seems to mean sufficient data to plot a yearly cycle. After rereading section 4.2.1 a few times I have come to the conclusion that the only new contributions are lines 482-486, with the rest either being obvious or previously known (Eicken 2002). I find is also very confusing that the authors do not mention more recent attempts at analyzing the salinity cycle. For example the authors cite Griewank and Notz 2015, but fail to mention that Griewank and Notz 2015 not only look at the same seasonal cycle of salinity, they even used the same ice core data from Alaska! I urge the authors to single out what their analysis provides that others can not, and properly frame their results in the context of what else has been achieved in the last decade. If the authors are not trying to provide a reference climatology they should avoid raising that expectation, and if they are they should provide and link to that data in some format that others can use.

Figures

1 Nice plot, no complaints.

2 Left subplot: red and green lines are not distinguishable by red-green colorblind people, the black dashed line is barely visible against the dark blue, why does the plot start and end so abruptly cutting off the ice core points. Right subplots: Far too many dots lie over each other obscuring what is happening. If it is important that the reader can distinguish the individual plots, make the figure big enough for this to be possible.

3 Too many lines lie over each other, with hard to distinguish colors (e.g. yellow

vs light green on white background). Image quality is poor, lines blur together when zoomed in. In subplot a the line farthest to the right seems to randomly switch from grey to orange to red and back to grey. The axis limits are poorly chosen. Temperature in b and d goes to -20 or so, but the lowest value is -11, in a) and b) salinity need only go to 11 or 12, subplots e,f,g,h, have the same issue. Using better x-axis limits would increase the distance between the individual lines, making it easier to tell them apart. Saving as a vector format would allow the reader to zoom in.

4 This figure has large amounts of redundant information and dead space. The lines which are interesting to compare to each other (e.g. salinity at 25-35 TDD from Van mijen Fjord vs Utqiagvik) are too far apart to compare easily. If only one core of data is present, while technically correct it seems misleading to label it as "max". I am not sure what data is important in the plot and what the authors are trying to convey. Perhaps this is a plot better suited as supplementary information.

5 See comments on Figure 2. Subplot c is nicely done, but has nothing in common with a and b and I would recommend treating it as it's own figure. The light blue line (0.47 from ice bottom) is hard to see.

6 This is again a very busy plot, and I struggle to find what is relevant to support what the authors are trying to convey. Like Figure 4, this feels more like supplementary information. The colorbar is maxed out in many errors hiding the values. A symmetrical log scale (e.g. matplotlib.colors.SymLogNorm for Python) might help. A more minor detail, but using different colorbars for temperature and salinity would make it easier to identify which plots are comparable to each other.

7 The actual data in the plot is very small and hard to see. Given that all 4 subplots share the same x axis it seems that stacking them vertically makes more sense

than horizontally, or not? And why are differences to cores shown as bars, while differences to mass balance sites are plotted through a scatter plot?

8 What are the dotted lines? What time period/ice depth to the X and + represent? It should be explicitly stated if the dashed or drawn through line is + or -.

9 I do not understand this plot, nor understand why it is relevant. A quick link to the relevant subsection in the figure caption could help.

---

## Referee Comment (RC2) · Anonymous Referee #2 · 18 May 2020

Review of:
**"Seasonal and interannual variability of sea-ice state variables: Observations and predictions for landfast ice in northern Alaska and Svalbard"**
by Marc Oggier, Hajo Eicken, Meibing Jin, Knut Høyland

This manuscript looks at a large number of ice cores sampled over 20 years to describe the seasonal and inter-annual variability of landfast ice temperature and salinity, in two different locations. They then evaluate the performance of the sea ice model CICE at reproducing these observations. The subject and content of the manuscript is well aligned with The Cryosphere Discussion, and I believe that it has potential for publication. In particular, the authors show that grouping the different cores in terms of Degree Days at the moment of sampling instead of dates improves the definition of the vertical profiles and clarify the climatology. They also show that aligning the cores from the ice bottom increases the accuracy in the measurements in the lower portion of the ice. I also think very relevant the discussion on how the field observation limitations can hinder model validation, especially for the modeling community.

However, the manuscript suffers from its lack of focus and clarity. It is overly long, and focuses too much on the validation of previous literature, or on unnecessary speculations on the presented data. In its current form, it is very difficult to extract the relevant information from the results.

For these reasons, I would consider this manuscript for publication but only after it is significantly re-written to address the following points.

General comments:

1- I believe that the manuscript is overly long, and needs to be better organized. For instance, much of the methods and results are repeated in the discussion. Most of the analysis could be reported in a much shorter and clearer manuscript.

2- The result section is difficult to follow, with too many details that feel a bit disorganized. I believe that the important points should be identified and better highlighted.

3- The discussion should focus on their contributions and less on the confirmation of previous literature. While these are sometimes worth mentioning, they are too heavily discussed, which bury their actual findings.

4- Many figures are too complicated or ill-described in the captions. It makes it hard to find the relevant information and to cross-validate what is described in the text. This is especially true for figures from the model section.

Specific comments:

Abstract: I believe that some important contributions (e.g. a new method for sampling the ice core layers) are missing in the abstract.

L14 and rest of manuscript: I would not refer to ice salinity and temperature as "ice properties".

L16 and below: The version of CICE should be specified.

L54-55 : Too many « ice properties ». It makes this statement vague and confusing.

L58-59 : This sentence is hard to follow.

L113: "*The ice growth season was overall shorter and warmer than at Utqiaġvik*" -> this belongs in the results section.

L135-140 : Was any cross-validation made between the thermistor string measurements and ice core temperature measurements ? If so, it would be interesting to quantify the accuracy of the internal ice temperature measurements from ice cores. I am wondering how the extraction and handling of the cores may be influencing the temperature readings?

L145-150 : Aren't the temperature readings point-measurements ? I am a bit confused on what this re-sampling means in terms of the temperature and salinity profiles. I think this could be clarified.

L154 : Has this DD method been done before? If not, I think that this is a very interesting contribution and the wording should be changed to highlight this.

Section 2.3: I would like some missing information to be added in this section.
-   Which version of CICE are you using?
-   There is no information on the snow layers
-   Is the dynamical component active, or turned off? If it is active, how did you determine (and define) whether the location is land-fast?

Section 3: This section is tedious to read and would benefit from being re-organised to avoid back-and-forth. This is especially true for section 3.2.

L 215: What is defined as the "median standard deviations? I am confused, as it implies a distribution of stds, which are themselves statistics of a distribution. This clarification is especially important given that there are many similar comments on this later in the manuscript.

L245 : This wording is too strong. We cannot determine the performance of a model at reproducing the trends and variability only from the envelope formed by the simulated extrema.

L246-252: Much of these observations are not presented.

L253-289: These paragraphs are difficult to follow, with a bit of back and forth between the different figures, general comments and details about the different layers. It should be re-organised to focus on the important points.

L290: "*inter-annual variability between observation and model*": This is strangely formulated. I think the term "time series of the differences" would be more appropriate.

L298-299: This discussion on the differences between the IMBs and the core measurement should be assessed earlier, in the observation section. E.g. with L135-140.

L304: I am guessing that you mean "snow thickness"

L305-311: A description of the heat capacity computation should be included in the method sections. This paragraph is also confusing and may be a few more lines would be useful to clarified this. I was not able to validate the information given in the text from the figure.

L313: Where is this porosity value coming from?

L314: I guess that you mean Figure 9a

L314: What do you define as "natural variability"? The "deviation from natural variability" is confusing to me. Do you rather mean "deviation from the observations"?

L315-326: A lot of the information presented in this paragraph is not shown. I believe that the information would be better conveyed with a figure showing the measured and modeled values of brine fraction and porosity.

L334-339: I am not sure that I am following this reasoning. The previous lines were sufficient, in my opinion.

L350: missing words: *with respect* "to the" *climatology*

L351-369: This would fit better in the method section.

L373-374: How do you quantify the brine loss and the layer in which it is important?

L380-386: This is interesting and suggests that ice core measurements are limited for model validation. This could be discussed earlier when the model bias is presented. I think it would have helped making sense of figure 6.

L393: missing words : […] *may be* "due to*?"* *differences in sample handling*.

L397: I would removed the *", know to be"*. It confused me on weather you were referring to your method or to something else.

L400-405: I would have loved to read this in the method section. If the authors have more information on these differences between ice cores and thermistor string measurements, it would very interesting to include, as it would provide a better idea of the possible temperature bias.

L419-424: This feels un-necessary and out of place in this section.

Section 4.1.2: The discussion on "acceptable differences" feel arbitrary and un-necessary given that it is barely mentioned later in the text. The rest of this section does not really describe uncertainties of the model, and more about the methods.

L458-459: This statement is in contradiction with the lines below, where you indicate that smaller growth rates lead to thinner ice in the Van Mijen Fjord, and also at L224-226 where the temperature gradients are smaller in the ice growth season in this location. The larger growth rates are again mentioned at L529. This needs to be clarified.

L480: This could be shorter, as this was already described in the previous section.

L509: The end of the sentence is missing.

L513-517: This should be explained the first time this feature is mentioned, at L235. Is the attribution of the 0 salinity measurements to freshwater underplating speculated, or was it corroborated by other observations?

L525: I think the "with respect to" is not the right expression… "considering"?

L532-542: Is this observed in your case or inferred from previous literature? This should be written in light of your results, or left out of the analysis. These speculations should be supported or related to the results, or left out of the analysis.

L543-546: The precipitation, snow depth or the presence in superimposed ice was not presented in the results for the Van Mijen Fjord. Is this observed or speculated?

L549-551: This is a very interesting and useful statement.

L560: This is not shown in Fig. 7, but I think that it would be very useful to add this information in a figure, as the snow depth in often mentioned in the analysis.

L574-575: It is unclear how land-fast ice is simulated in the model. This is an important point to cover in the method section. How was "land-fast ice" defined in the model and was it confirmed that the grid location remained land-fast during the observation periods?

L592-594: It was previously mentioned that the salinity could also be underestimated by the drainage of brine even in the top layers (L480). Can it also be related to this?

Figure 3: error in the labels (max in red, max in blue). The max and min lines are also difficult to see.

Figure 4: Too much text in the figure, it is hard to spot the a), b) c).

Figure 5: Error bars are difficult to see

Figure 6: Also too much text in the figure. If the upper half section of the cores are hatched when the sampling is from the ice bottom (bottom panels), why not doing the same for the lower-half sections of the cores when sampled from the top (top panels)?

Figure 7: These plots are very small. I think that showing differences without known the actual value is difficult to judge whether these anomalies are important or not.

Figure 8: I do not understand this figure…

Figure 9: I also have difficulty understanding this figure, but I suspect that I am mostly confused by the phrasing in the caption: if the plots are showing the actual values, not the differences. What means "as a function of the model bias?"

---

## Author Comment (AC1) · 27 Jun 2020

We sincerely thank both reviewers for the constructive comments and time devoted to discussing our manuscript. We agree with the general comments from both reviewers that the manuscript is overly long, and requires reorganization. We plan to modify our manuscript in response to the comments as detailed below. We will clarify and enhance text describing the aims and scientific goals of this paper, which are:

1. In this paper, we develop a new framework to analyze sea ice core profile data sets. We introduce (i) a dual coordinate system referencing both the snow/ice and ice/ocean interface to capture relevant processes in both upper and lower ice

layers, and (ii) cumulative degree days (DD) as temporal reference to determine the mean, range and standard deviation of ice core profile data (i.e. a climatology of profile data).

2. We build a sea ice climatology based on existing collections of ice cores, and provide a readily available reference climatology, which will be available online in accordance with The Cryosphere requirements. Such climatology serves two goals. First, it may function as a model benchmarking tool to be used by the modeling community. To date, ice core data from Utqiagvik have been used in many scientific studies (e.g., Griewank and Notz, 2015; Petrich et al., 2013; Vancoppenolle, 2007), but have lacked a common framework for analysis and validation, making intercomparisons and quantitative assessments of model performance difficult. The framework developed here can also support sampling strategies and modeling of biogeochemical processes in and under sea ice (Steiner et al., 2016). Second, such climatology can be used to evaluate representativeness and guide growth of artificial sea ice in a range of laboratory experiments, such as in the oil-in-ice experiments of Pegau et al. (2016) or Petrich et al. (2018).We investigate the climatology in terms of mean down-core profiles and variability, examine the sources of variability (spatial, seasonal, methodological), and compare our findings with results from other published studies.

3. We focus on the benefits of the developed framework to evaluate the performance of the CICE Los Alamos sea ice model in replicating key aspects of ice core climatology, and the implications for studies of sea ice processes. The choice of the model was motivated by its wide adoption in climate system models, as well as ocean and weather forecasting models.

In order to improve the readability of the paper, we propose to restructure the manuscript based on these goals. A revised outline of our manuscript is attached as an appendix.

[Figure]

*In the submitted manuscript Oggier et al. have analyzed 180 fast-ice cores from Alaska and 60 ice cores from Svalbard gathered over roughly a decade. The cores are binned together by degree days (a unit the authors use instead of time to sort the cores into differing stages of the sea-ice life cycle), and various properties of the ice are discussed in regards to the sea-ice's life cycle and how much they vary from year to year. At both study location simulations are run using the 1D CICE sea ice model, and the model output is compared to the ice core data and other measurements taken from the many measurement excursions over the years.*

*Given that the paper discusses sea ice in detail, it falls within the scope of TC. The novelty of the paper lies less in the data and simulations used, and more in the methods used to compare sea ice from differing times and of different thickness. The many cores in addition to the model simulations provide the authors with a wealth of data to draw conclusions from. However, I find that the authors struggle to distil new insights from this wealth of data. A lack of clear scientific questions made it difficult to judge if the methods used are suitable, and neither the introduction nor the structure of the paper give the reader a sufficient frame of reference to follow. I am unable to distinguish when the authors summarize what has already previously been known from when the authors are introducing their own results.*

*In addition to the missing storylines and poor flow of the paper, the figures of the manuscript are extremely busy and difficult to process. The colors chosen are difficult to distinguish and not colorblind friendly, and data is often obscured by overlapping lines/dots. A further issue is that the authors do not follow the TC data policy. I found no statements regarding the availability of the data used, nor a link or reference to the precise model version of CICE used to run the simulations.*

RE: We hope the proposed restructuring of the manuscript outlined above addresses those general comments.

*For the reasons listed, I recommend that the paper be rejected. However, since the*

[Figure]

*data itself is solid and because there are many interesting facts scattered throughout the submitted manuscript, I strongly encourage the authors to refine the aims and scope of the manuscript and then resubmit. My impression of the submitted manuscript is that it attempts to cover too many things at once.*

*The remainder of the review will raise some general issues I found particularly problematic, followed by detailed comments on the individual figures.*

RE: In the following section, reviewer's comments are shown in italic.

**General issues**

Missing questions

*Currently, the paper introduction raises no questions. It simply states that data is needed, and that the authors provide data. If this is the case this paper should be reformulated as a technical description or data paper. There are a wealth of questions that could be raised. Here just two examples:*

- *The decade long collection of ice cores in Alaska is unique in the number of cores gathered and the time covered. However, it is unclear if the data contains additional variability in addition to the interseasonal and spatial variability due to the constantly changing participants who extracted the cores. In this paper we .....*

- *Reference profiles of salinity are commonly provided in normalized coordinates from ice-snow interface to ice-ocean interface (e.g. lots of citations). This approach functions poorly for first year ice which changes thickness rapidly. In this paper we will determine if providing reference profiles in meters from the ice-ocean interface is more suitable for studies of ice biogeochemistry.*

RE: We agree we did not clearly identify the goals of the paper in the introduction. In the revised introduction we will highlight the four goals, previously identified, namely:

1. Introducing a new framework to analyze ice core profile data based on (i) a dual coordinate system referencing both the snow/ice and ice/ocean interface, and (ii) a DD as temporal reference.

2. Building a climatology based on the decade-long ice core collection from Utqi-agvik which has been used in many scientific studies (e.g., Griewank and Notz, 2015; Petrich et al., 2013; Vancoppenolle, 2007), but has lacked a common framework for such analysis and validation. In this paper, we determined that the cumulative DD at the date of coring, rather than the date of coring are most appropriate for intercomparison and temporal classification. Also, ice core profile data are analyzed both with respect to the ice-snow and ice-ocean interface, an approach that is found to be more appropriate to aggregate and intercompare ice core profile data.

3. Investigating the climatology in terms of mean down-core profiles and variability. While the seasonal evolution of salinity profiles is well described (Malmgren, 1927; Petrich and Eicken, 2017), and the spatial variability for salinity is commonly expected to vary within 0.5 to 1 ‰; most analyses are based on single season data sets (Weeks and Lee, 1962; Eicken et al. 2002; Gough et al. 2012). In this paper we focus on two-decade-long data set, which allows us to quantify the interannual variability, and examine the sources of variability (spatial, seasonal, methodological).

4. Evaluating the CICE Los Alamos model, integrated in many coupled climate system models. Turner et al.

Model-observation comparison

*The authors failed to convey what is gained by including the model in this paper. The inclusion of the model is further complicated by the authors not cleanly separating what is used to force/tune the model versus what is used to evaluate it. The ocean heat flux was tuned to fit the ice depth, and then the ice thickness was used to evaluate the model performance? It has been known since the 60s that ice thickness is dominated by ocean heat flux, atmospheric heat flux, and snow depth. Accordingly evaluating simulated ice depth says less about the model than the forcing data. Are the authors attempting to verify the consistency of the forcing data?*

RE: This is a very helpful comment. We will build on the reviewer's comment to revise the paper. We thank the reviewer for pointing out the circular reasoning between the use of ice thickness to tune the ocean heat flux, and subsequent use of the ice thickness to evaluate the model performance. As stated later, we wanted to verify the model consistency. Although the ocean heat flux is adjusted to the observed ice thickness, but it is the same seasonal cycle for every year, so the modeled ice thickness still includes interannual variations independent of the ocean heat flux. Thus validation in Fig 7 is not affected by the use of ocean heat flux.

*All aspects of the model are studied in much greater detail in other papers, for example Lecomte et al. 2013 in regards to snow. Are they attempting to evaluate the salinity parametrizations? If so they should refer to and frame their results in regards to recent research in that area, e.g. Max Thomas et al. 2020 "Tracer Measurements in ....", or Jacob Buffo et al. 2018 "Multiphase reactive transport and platelet Ice Accretion ..." developing*

RE: The CICE Los Alamos model has been widely adopted by climate system modelers, and claims to replicate desalination with the added mushy thermodynamic component (Turner et al., 2013). Knowing how the model replicates interannual and seasonal variability in terms of salinity and temperature variables is key for research using those

variables as proxies for further modeling (e.g., contaminant transport, biogeochemistry). To our knowledge, this paper is the first to evaluate the model performance with decade-long observation recordsFollowing the reviewer's pointer, we will include a comparison of the model performance to previous model assessments.

*Similarly, why are the authors looking at ice heat capacity during melting? It is already known from basic sea-ice thermodynamics that the heat capacity is very sensitive to changes in salinity and temperature close to the melting point, no model or observations are needed to confirm this. The heat capacity also has very little impact on simulating ice melt compared to the completely dominating effect of the surface albedo. I personally found the modelling aspect of the paper very unconvincing, and would encourage the authors to figure out exactly how the model helps them convey their results.*

RE: We agree, and hence we will take the reviewer guidance and remove this section, including Figure 8 and 9.

Structure

*I would highly recommend that the authors rethink their current approach of having one big results section, followed by a very long discussion section. It is also not helpful that the current results section is predominantly filled with descriptions of figures. By just describing data in "results" without a purpose the reader has no guidance what is important. And then when the authors raise points in the discussion many pages later the reader has already long forgotten what they saw. I recommend going through the scientific questions one by one, and supply the specific data needed to address each specific question as it is being discussed.*

RE: Following the reviewer's comment, we will restructure the manuscript. The current results and discussion sections will be reorganized into three much tighter sections.

Each one will address one of the highlighted goals. The section titles of the restructured paper follow below, with more details provided in the appendix:

1. Introduction

2. Methods

3. A spatial and temporal reference framework for analysis of ice core profile data sets

4. Interannual and seasonal variability

5. Comparison with CICE model output

*Despite how often it is referred to I do not know what the authors mean by climatology. It initially sounds like they are attempting to provide a reference set of profiles for others to use, similar to a sea surface temperature climatology map. But by the time we have reached section 4.2., "climatology" seems to mean sufficient data to plot a yearly cycle.*

RE: The first goal of paper is to develop a new framework to aggregate ice core data, and propose a common framework for both model comparisons, and evaluate the representativeness of artificially grown sea ice by comparing it with data from a specific region, here Alaska and Svalbard. However, in contrast with, e.g., the availability and coverage of sea surface temperature data to produce a surface temperature climatology map, ice core data coverage is typically more limited and requires aggregation as discussed in the paper.

*After rereading section 4.2.1 a few times I have come to the conclusion that the only new contributions are lines 482-486, with the rest either being obvious or previously known (Eicken 2002).*

RE: While the evolution of sea-ice salinity profiles is well described in the literature (Petrich and Eicken, 2017, Gough et al. 2012, Eicken et al. 2002, Weeks and Lee,

1962), the published studies are based on single season and single location data sets. In this paper, we analyze larger data sets, spreading across a longer period time. In addition, we developed a new analysis framework (dual interface reference horizons, DD classification of profile) that allows us to examine spatial and seasonal evolution, but also interannual variability, the latter lacking in previous studies. Finally, we included in our analysis two contrasting locations to examine variability in a broader context.

*I find is also very confusing that the authors do not mention more recent attempts at analyzing the salinity cycle. For example the authors cite Griewank and Notz 2015, but fail to mention that Griewank and Notz 2015 not only look at the same seasonal cycle of salinity, they even used the same ice core data from Alaska! I urge the authors to single out what their analysis provides that others can not, and properly frame their results in the context of what else has been achieved in the last decade.*

RE: We appreciate the pointer to the use of the same ice core data from Utqiagvik by Griewank and Notz (2015). The motivation of our current study to develop a common and consistent framework for ice core data is in line with Griewank and Notz (2015) findings, and we will use the paper to illustrate the importance of such framework. Through curation of a broader range of data collected at Utqiagvik, the data set analyzed here is almost double the size of that used by Griewank and Notz (2015). In addition, as we were able to relate each ice core to the date of ice formation of landfast ice, we were able to directly compare the observational data to model output, with the model forced with reanalysis data for that same location. Griewank and Notz (2015) compared the observations to simulations forced at nine locations over the Arctic, which they recognized as being less than ideal. Besides, the introduction of a new method to analyze sea ice profiles, that does not require normalization of profile depth, which disproportionately stretches the upper and lower high salinity layer of thinner ice (January), relatively to thicker ice (March), allows for greater accuracy in the mean and standard deviation of salinity and temperature profiles.

*If the authors are not trying to provide a reference climatology, they should avoid raising*

*that expectation, and if they are they should provide and link to that data in some format that others can use.*

RE: We will make the climatology data accessible online as part of the supplemental materials, so that readers have direct access. The ice core data sets are already posted at the Arctic Data Center (Utqiagvik, https://doi.org/10.18739/A2XP6V39R) or Zenodo (Van Mijen Fjord, https://doi.org/10.5281/zenodo.3737133).

Figures:

RE: In general, we will improve the figure using a colorblind friendly color map (e.g. cividis or viridis)

*1 Nice plot, no complaints.* RE: Thanks.

*2 Left subplot: red and green lines are not distinguishable by red-green colorblind people, the black dashed line is barely visible against the dark blue, why does the plot start and end so abruptly cutting off the ice core points. Right subplots: Far too many dots lie over each other obscuring what is happening. If it is important that the reader can distinguish the individual plots, make the figure big enough for this to be possible.*

RE: On the left subplot, we will choose more appropriate colors for colorblind people, and modify the range of the x-axis in order not to cut the ice core points. For the right subplots, we will remove the color reference to the year, as we do not make use of this information in the paper.

*3 Too many lines lie over each other, with hard to distinguish colors (e.g. yellow vs light green on white background). Image quality is poor, lines blur together when zoomed in. In subplot a the line farthest to the right seems to randomly switch from grey to orange to red and back to grey. The axis limits are poorly chosen. Temperature in b and d goes to -20 or so, but the lowest value is -11, in a) and b) salinity need only go to 11 or 12, subplots e,f,g,h, have the same issue. Using better x-axis limits would increase the*

*distance between the individual lines, making it easier to tell them apart. Saving as a vector format would allow the reader to zoom in.*

RE: We will correct the labels, and choose better axis labels, especially for the temperature.

*4 This figure has large amounts of redundant information and dead space. The lines which are interesting to compare to each other (e.g. salinity at 25-35 TDD from Van Mijen Fjord vs Utqiagvik) are too far apart to compare easily. If only one core of data is present, while technically correct it seems misleading to label it as "max". I am not sure what data is important in the plot and what the authors are trying to convey. Perhaps this is a plot better suited as supplementary information.*

RE: We will simplify the label of the y-axis in order to highlight a, b, c, and try to remove redundant information and dead space. If data from only one core is presented, we will remove the min/max values, and display only a black line. This figure not only supports the developed framework to generate a climatology, but also highlights the limitation of data scarcity in the process. The full climatology is available as supplemental material, and the data for temperature and salinity profiles will be made available online.

*5 See comments on Figure 2. Subplot c is nicely done, but has nothing in common with a and b and I would recommend treating it as its own figure. The light blue line (0.47 from ice bottom) is hard to see.*

RE: We will split this figure into two figures, with the subplot c) treated as its own figure, which will allow us to increase the error bars.

*6 This is again a very busy plot, and I struggle to find what is relevant to support what the authors are trying to convey. Like Figure 4, this feels more like supplementary information. The colorbar is maxed out in many errors hiding the values. A symmetrical log scale (e.g. matplotlib.colors.SymLogNorm for Python) might help. A more minor detail, but using different colorbars for temperature and salinity would make it easier to*

*identify which plots are comparable to each other.*

RE: We will keep the figure in the manuscript, as we feel it is important to present the difference between the modeled and observed seasonal evolution. However, we will consider the suggestion of using a symmetrical log scale, and different colormaps for temperature and salinity.

*7 The actual data in the plot is very small and hard to see. Given that all 4 subplots share the same x axis it seems that stacking them vertically makes more sense than horizontally, or not? And why are differences to cores shown as bars, while differences to mass balance sites are plotted through a scatter plot?*

RE: We will increase the size of the plot, and stack them vertically as they all share the same x-axis. This plot will be presented as supplemental material

*8 What are the dotted lines? What time period/ice depth to the X and + represent? It should be explicitly stated if the dashed or drawn through line is + or -.*

RE: Following the reorganization and shortening of the manuscript, we will remove this figure

*9 I do not understand this plot, nor understand why it is relevant. A quick link to the relevant subsection in the figure caption could help.*

RE: Following the reorganization and shortening of the manuscript, we will remove this figure

Please also note the supplement to this comment:
https://tc.copernicus.org/preprints/tc-2020-52/tc-2020-52-AC1-supplement.pdf

[Figure]

**Supplement:**

**Revised outline of manuscript**

*In the following outline, we retained the figure numbering scheme of the manuscript.*

1. Introduction
2. Methods
   2.1. Study sites
      2.1.1. Utqiaġvik, Alaska
      2.1.2. Van Mijen Fjord
   2.2. Field measurements, sample and data processing
   2.3. Aggregation of ice core profile data sets
      2.3.1. Degree day classification
      2.3.2. Profile depth referenced to snow/ice and ice/ocean interface
   2.4. Model simulations
3. A spatial and temporal reference framework for analysis of ice core profile data sets
   3.1. Results
      New figure: Difference in classification by using DD vs date of coring or ice thickness for aggregating ice core data.
      Figure 3: Present ice core profile datasets with dual reference horizons (sea ice surface and bottom).
   3.2. Discussion: Advantages of the proposed framework
      We will discuss the benefits of using a DD framework and dual coordinate system referencing both the snow/ice and ice/ocean interface. We will discuss requirements in terms of sampling frequency, and amount of ice cores collected for each sampling event to produce a climatology (mean, variance, extremes of profile data) that meets the broader aims outlined in the paper.
4. Interannual and seasonal variability
   4.1. Results
      Figure 4: We will describe the shape of core profiles and their variability. We will highlight the difference between previous observations based on a single season (e.g., Eicken et al., 2002; Gough et al., 2012), and this long-term data set.
   4.2. Discussion
      4.2.1. Source of errors
         We will discuss sources of error (temperature bias, brine drainage), as well as inconsistencies or deviations from core sampling and processing protocols.
      4.2.2. Seasonal evolution and interannual variability
         We will discuss the environmental processes responsible for the observed spatial and seasonal variations in ice temperature and salinity.
5. Comparison with CICE model output
      Results
      Figure 5a, b: comparing modeled and observed ice and snow thickness to check the consistency of model tuning.
      Figure 7a, c, d: differences between model observation. This figure will be moved into supplemental materials.

Figure 6: Seasonal anomaly between model and observation: we will describe the success (absence of brine loss and bias in temperature measurement), and failure of the model (desalination during the melt season with more detail in Figure 5c).

Figure 5c: failure of model to capture desalination during the melt season

5.1. Discussion

5.1.1. Sources of error

We discuss the model calibration in terms of ice and snow thicknesses, and limitations of the stand-alone mode we used.

5.1.2. Seasonal evolution and interannual variability

As the model does not reproduce small-scale spatial variability, and does not capture errors due to the sampling method, we discuss the benefits  of the model in terms of reproducing seasonal evolution and interannual variability of salinity and temperature profiles and the implications for studies of sea ice processes.

6. Conclusions

---

## Author Comment (AC2) · 27 Jun 2020

We sincerely thank both reviewers for the constructive comments and time devoted to discussing our manuscript. We agree with the general comments from both reviewers that the manuscript is overly long, and requires reorganization. We plan to modify our manuscript in response to the comments as detailed below. We will clarify and enhance text describing the aims and scientific goals of this paper, which are:

1. In this paper, we develop a new framework to analyze sea ice core profile data sets. We introduce (i) a dual coordinate system referencing both the snow/ice and ice/ocean interface to capture relevant processes in both upper and lower ice

layers, and (ii) cumulative degree days (DD) as temporal reference to determine the mean, range and standard deviation of ice core profile data (i.e. a climatology of profile data).

2. We build a sea ice climatology based on existing collections of ice cores, and provide a readily available reference climatology, which will be available online in accordance with The Cryosphere requirements. Such climatology serves two goals. First, it may function as a model benchmarking tool to be used by the modeling community. To date, ice core data from UtqiaÄąvik have been used in many scientific studies (e.g., Griewank and Notz, 2015; Petrich et al., 2013; Vancoppenolle, 2007), but have lacked a common framework for analysis and validation, making intercomparisons and quantitative assessments of model performance difficult. The framework developed here can also support sampling strategies and modeling of biogeochemical processes in and under sea ice (Steiner et al., 2016). Second, such climatology can be used to evaluate representativeness and guide growth of artificial sea ice in a range of laboratory experiments, such as in the oil-in-ice experiments of Pegau et al. (2016) or Petrich et al. (2018).We investigate the climatology in terms of mean down-core profiles and variability, examine the sources of variability (spatial, seasonal, methodological), and compare our findings with results from other published studies.

3. We focus on the benefits of the developed framework to evaluate the performance of the CICE Los Alamos sea ice model in replicating key aspects of ice core climatology, and the implications for studies of sea ice processes. The choice of the model was motivated by its wide adoption in climate system models, as well as ocean and weather forecasting models.

In order to improve the readability of the paper, we propose to restructure the manuscript based on these goals. A revised outline of our manuscript is attached as an appendix.

In the following section, reviewer's comments are shown in italic.

*In the submitted manuscript Oggier et al. have analyzed 180 fast-ice cores from Alaska and 60 ice cores from Svalbard gathered over roughly a decade. The cores are binned together by degree days (a unit the authors use instead of time to sort the cores into differing stages of the sea-ice life cycle), and various properties of the ice are discussed in regards to the sea-ice's life cycle and how much they vary from year to year. At both study location simulations are run using the 1D CICE sea ice model, and the model output is compared to the ice core data and other measurements taken from the many measurement excursions over the years.*

*Given that the paper discusses sea ice in detail, it falls within the scope of TC. The novelty of the paper lies less in the data and simulations used, and more in the methods used to compare sea ice from differing times and of different thickness. The many cores in addition to the model simulations provide the authors with a wealth of data to draw conclusions from. However, I find that the authors struggle to distil new insights from this wealth of data. A lack of clear scientific questions made it difficult to judge if the methods used are suitable, and neither the introduction nor the structure of the paper give the reader a sufficient frame of reference to follow. I am unable to distinguish when the authors summarize what has already previously been known from when the authors are introducing their own results.*

*In addition to the missing storylines and poor flow of the paper, the figures of the manuscript are extremely busy and difficult to process. The colors chosen are difficult to distinguish and not colorblind friendly, and data is often obscured by overlapping lines/dots. A further issue is that the authors do not follow the TC data policy. I found no statements regarding the availability of the data used, nor a link or reference to the precise model version of CICE used to run the simulations.*

RE: We hope the proposed restructuring of the manuscript outlined above addresses those general comments.

*For the reasons listed, I recommend that the paper be rejected. However, since the data itself is solid and because there are many interesting facts scattered throughout the submitted manuscript, I strongly encourage the authors to refine the aims and scope of the manuscript and then resubmit. My impression of the submitted manuscript is that it attempts to cover too many things at once.*

*The remainder of the review will raise some general issues I found particularly problematic, followed by detailed comments on the individual figures.*

RE: In the following section, reviewer's comments are shown in italic.

*In the submitted manuscript Oggier et al. have analyzed 180 fast-ice cores from Alaska and 60 ice cores from Svalbard gathered over roughly a decade. The cores are binned together by degree days (a unit the authors use instead of time to sort the cores into differing stages of the sea-ice life cycle), and various properties of the ice are discussed in regards to the sea-ice's life cycle and how much they vary from year to year. At both study locations simulations are run using the 1D CICE sea ice model, and the model output is compared to the ice core data and other measurements taken from the many measurement excursions over the years.*

*Given that the paper discusses sea ice in detail, it definitely falls within the scope of TC. The novelty of the paper lies less in the data and simulations used, and more in the methods used to compare sea ice from differing times and of different thickness. The many cores in addition to the model simulations provide the authors with a wealth of data to draw conclusions from. However, I find that the authors struggle to distil new insights from this wealth of data. A lack of clear scientific questions made it difficult to judge if the methods used are suitable, and neither the introduction nor the structure of the paper give the reader a sufficient frame of reference to follow. I am unable to distinguish when the authors summarize what has already previously been known from when the authors are introducing their own results.*

*In addition to the missing storylines and poor flow of the paper, the figures of the*

[Figure]

*manuscript are extremely busy and difficult to process. The colors chosen are difficult to distinguish and not colorblind friendly, and data is often obscured by overlapping lines/dots. A further issue is that the authors do not follow the TC data policy. I found no statements regarding the availability of the data used, nor a link or reference to the precise model version of CICE used to run the simulations.*

RE: We hope the proposed restructuring of the manuscript outlined above addresses those general comments.

*For the reasons listed, I recommend that the paper be rejected. However, since the data itself is solid and because there are many interesting facts scattered throughout the submitted manuscript, I strongly encourage the authors to refine the aims and scope of the manuscript and then resubmit. My impression of the submitted manuscript is that it attempts to cover too many things at once.*

*The remainder of the review will raise some general issues I found particularly problematic, followed by detailed comments on the individual figures.*

RE: In the following section, reviewer's comments are shown in italic.

**General comments:**

1. *I believe that the manuscript is overly long, and needs to be better organized. For instance, much of the methods and results are repeated in the discussion. Most of the analysis could be reported in a much shorter and clearer manuscript.*

2. *The result section is difficult to follow, with too many details that feel a bit disorganized. I believe that the important points should be identified and better highlighted.*

*3. The discussion should focus on their contributions and less on the confirmation of previous literature. While these are sometimes worth mentioning, they are too heavily discussed, which bury their actual findings.*

3. Response (RE.): We hope the proposed restructuring of the manuscript outlined above addresses the first 3 general comments.

*4. Many figures are too complicated or ill-described in the captions. It makes it hard to find the relevant information and to cross-validate what is described in the text. This is especially true for figures from the model section.*

RE.: We will rework the figures based on specific guidance, shorten the axis label, and use colorblind friendly coloring palettes. We will also improve the captions. Some figures (8 and 9) will be eliminated to tighten the manuscript based on guidance provided.

**Specific comments:**

*Abstract: I believe that some important contributions (e.g. a new method for sampling the ice core layers) are missing in the abstract. L14 and rest of manuscript: I would not refer to ice salinity and temperature as ice properties.*

RE: We will revise the abstract to include the new method for sampling ice core. Throughout the text we will use explicitly the terms salinity and temperature rather than ice properties.

*L16 and below: The version of CICE should be specified.*

RE: Thanks for catching that. We will add the version no. (6).

*L54-55 : Too many Âń ice properties Âż. It makes this statement vague and confusing.*

RE: Throughout the text we will use explicitly the terms salinity and temperature rather

than ice properties.

*L58-59 : This sentence is hard to follow.*

RE. In the revised manuscript, we will reformulate the sentence.

*L113: "The ice growth season was overall shorter and warmer than at UtqiaÄąvik" -> this belongs in the results section.*

RE. Thanks for the comment, we will move it to the results.

*L135-140 : Was any cross-validation made between the thermistor string measurements and ice core temperature measurements? If so, it would be interesting to quantify the accuracy of the internal ice temperature measurements from ice cores. I am wondering how the extraction and handling of the cores may be influencing the temperature readings?*

RE: We did some cross-validation, but we prefer not to include to keep the manuscript shorter. We propose to add a couple of sentences about this in a "sources of error" subsection within the climatology section in our reorganized manuscript.

*L145-150 : Aren't the temperature readings point measurements? I am a bit confused on what this re-sampling means in terms of the temperature and salinity profiles. I think this could be clarified.*

RE: We will clarify. Salinity and temperature are point measurements. Depths of temperature measurement, and salinity section depths are not homogeneous throughout all cores. Thus we had to re-sample both salinity and temperature profiles in order to compare them all together.

*L154 : Has this DD method been done before? If not, I think that this is a very interesting contribution and the wording should be changed to highlight this.*

RE: DD is often used to estimate sea ice growth, however, to our knowledge it has not been used to classify ice cores prior.

*Section 2.3: I would like some missing information to be added in this section.*

RE: we will update the methods section.

*- Which version of CICE are you using?*

RE: Thanks for catching that. We will indicate that we used the developer's version of CICE 5, which allows a standalone mode, while CICE 6 can only be run in the global domain. There is no difference in the parametrization of the mushy layer thermodynamics between both versions.

*- There is no information on the snow layers* RE: The snow cover is treated as a single layer. We will specify this in the method.

*- Is the dynamical component active, or turned off? If it is active, how did you determine (and define) whether the location is land-fast?*

RE: The dynamical componentis active. We define the onset of formation of landfast sea ice as the onset of formation of ice which persist until the start of spring melt. The CICE standalone mode was designed for no horizontal ice motion, so it resembles landfast sea ice with no ice advection.

*Section 3: This section is tedious to read and would benefit from being reorganized to avoid back-and-forth. This is especially true for section 3.2.*

RE: As stated earlier, we will reorganize this section.

*L 215: What is defined as the "median standard deviations? I am confused, as it implies a distribution of stds, which are themselves statistics of a distribution. This clarification is especially important given that there are many similar comments on this later in the manuscript.*

RE: Thanks for the comment. We will clarify in the revised manuscript. By median standard deviation, we refer to the median value of all the standard deviations within a DD interval.

*L245 : This wording is too strong. We cannot determine the performance of a model at reproducing the trends and variability only from the envelope formed by the simulated extrema.*

RE: We will reformulate in the revised manuscript.

*L246-252: Much of these observations are not presented.*

RE: Figure 5a displays the modeled vs observed ice thickness, while Figure 5b displays the modeled vs observed snow thickness. In the figures, we display the correlation coefficient for both locations, and gives the correlation coefficient at each location in the plot.

*L253-289: These paragraphs are difficult to follow, with a bit of back and forth between the different figures, general comments and details about the different layers. It should be reorganized to focus on the important points.*

RE: We will reorganize and tighten the text and focus on the more important points.

*L290: "inter-annual variability between observation and model": This is strangely formulated. I think the term "time series of the differences" would be more appropriate.*

RE: Thanks for the proposed formulation

*L298-299: This discussion on the differences between the IMBs and the core measurement should be assessed earlier, in the observation section. E.g. with L135-140.*

RE: We will discuss this earlier in the observation section.

*L304: I am guessing that you mean "snow thickness"*

RE: We thank the reviewer for pointing this out.

*L305-311: A description of the heat capacity computation should be included in the method sections. This paragraph is also confusing and may be a few more lines would be useful to clarify this. I was not able to validate the information given in the text from*

*the figure.*

RE: Since we will remove Figure 8, this will be no longer needed.

*L313: Where is this porosity value coming from?*

RE: We will add text on how porosity was computed according to Cox and Weeks' formula.

*L314: I guess that you mean Figure 9a*

RE: We thank the reviewer for pointing this out.

*L314: What do you define as "natural variability"? The "deviation from natural variability" is confusing to me. Do you rather mean "deviation from the observations"?*

RE: Yes, we mean the deviation from the observation.

*L315-326: A lot of the information presented in this paragraph is not shown. I believe that the information would be better conveyed with a figure showing the measured and modeled values of brine fractions and porosity.*

RE: This information is contained in Figure 9, but the figure itself as pointed out is difficult to interpret. In response to guidance on tightening the text, we will remove this section altogether.

*L334-339: I am not sure that I am following this reasoning. The previous lines were sufficient, in my opinion.*

RE: Thanks. We will remove those lines.

*L350: missing words: with respect "to the" climatology*

RE: We thank the reviewer to point this out.

*L351-369: This would fit better in the method section.*

RE: We will consider this.

*L373-374: How do you quantify the brine loss and the layer in which it is important?*

RE: We look at the difference between model and observation. In addition, we compute the brine volume fraction using Cox and Weeks' formula, and found that the porosity in the lower 10 cm of the ice cover is above 5%. This value corresponds to the porosity threshold proposed by Golden et al. (2007) which corresponds to a permeability allowing vertical brine movement. The main caveat is that bulk porosity is computed from temperature and bulk salinity.

*L380-386: This is interesting and suggests that ice core measurements are limited for model validation. This could be discussed earlier when the model bias is presented. I think it would have helped making sense of figure 6.*

RE: This is a good idea. We will discuss this at the end of our climatology section in our reorganized manuscript.

*L393: missing words : [...] may be "due to?" differences in sample handling.*

RE: We thank the reviewer for pointing this out.

*L397: I would removed the ", know to be". It confused me on weather you were referring to your method or to something else.*

RE: Yes. Good suggestion.

*L400-405: I would have loved to read this in the method section. If the authors have more information on these differences between ice cores and thermistor string measurements, it would very interesting to include, as it would provide a better idea of the possible temperature bias.*

RE: We propose to move this comment to a "sources of error" subsection within the climatology section in our reorganized manuscript.

*L419-424: This feels un-necessary and out of place in this section.*

RE: We will remove this text.

*Section 4.1.2: The discussion on "acceptable differences" feel arbitrary and unnecessary given that it is barely mentioned later in the text. The rest of this section does not really describe uncertainties of the model, and more about the methods.*

RE: We will remove this text.

*L458-459: This statement is in contradiction with the lines below, where you indicate that smaller growth rates lead to thinner ice in the Van Mijen Fjord, and also at L224-226 where the temperature gradients are smaller in the ice growth season in this location. The larger growth rates are again mentioned at L529. This needs to be clarified.*

RE: Yes, we will clarify in the revised document

*L480: This could be shorter, as this was already described in the previous section.*

RE: Yes, we will shorten this section.

*L509: The end of the sentence is missing.*

RE: Thanks, we will correct.

*L513-517: This should be explained the first time this feature is mentioned, at L235. Is the attribution of the 0 salinity measurements to freshwater underplating speculated, or was it corroborated by other observations?*

RE: We will bring this up this earlier on in the text. Limited under ice salinity measurements made at the sampling site during some of the sampling campaigns indicates that in spring freshwater underplating is common at this location.

*L525: I think the "with respect to" is not the right expression... "considering"?*

RE: We will follow your suggestion.

*L532-542: Is this observed in your case or inferred from previous literature? This should be written in light of your results, or left out of the analysis. These speculations*

*should be supported or related to the results, or left out of the analysis.*

RE: Both snow ice and superimposed ice have been observed during the year the ice cores have been taken. However, since we observed preferentially a low-salinity layer at the surface, we speculate that superimposed ice is more common, and propose an explanation.

*L543-546: The precipitation, snow depth or the presence in superimposed ice was not presented in the results for the Van Mijen Fjord. Is this observed or speculated?*

RE: Superimposed ice and snow ice formation is very common in Van Mijen Fjord (Hoyland, 2009), and depends on local weather conditions. We decided not to include a detailed analysis of it.

*L549-551: This is a very interesting and useful statement.*

RE: Thanks.

*L560: This is not shown in Fig. 7, but I think that it would be very useful to add this information in a figure, as the snow depth in often mentioned in the analysis.*

RE: Difference between modeled and observed snow depth at maximum ice thickness is shown in Figure 7c. We will add the subplot number.

*L574-575: It is unclear how land-fast ice is simulated in the model. This is an important point to cover in the method section. How was "land-fast ice" defined in the model and was it confirmed that the grid location remained land-fast during the observation periods?*

RE: We will clarify that in the methods section. We run the CICE model on a single grid cell.

*L592-594: It was previously mentioned that the salinity could also be underestimated by the drainage of brine even in the top layers (L480). Can it also be related to this?*

RE: We do not think so. Salinity in the upper layer could be underestimated by brine drainage only in thin ice (< 30 cm), in which the porosity remains high throughout the whole ice cover.

**Figures:**

RE: In general, we will improve the figure using a colorblind friendly color map (e.g. cividis or viridis)

*Figure 3: error in the labels (max in red, max in blue). The max and min lines are also difficult to see.*

RE: We will correct the labels, and choose better axis labels, especially for the temperature.

*Figure 4: Too much text in the figure, it is hard to spot the a), b) c).* RE: We will simplify the label of the y-axis in order to highlight a, b, c, and try to remove redundant information and dead space. If data from only one core is presented, we will remove the min/max values, and display only a black line.

*Figure 5: Error bars are difficult to see* RE: We will split this figure into two figures, with the subplot c) treated as its own figure, which will allow us to increase the error bars.

*Figure 6: Also too much text in the figure. If the upper half section of the cores are hatched when the sampling is from the ice bottom (bottom panels), why not doing the same for the lower-half sections of the cores when sampled from the top (top panels)?*

RE: We will follow the suggestion, and only present the upper-half sections of the ice core in the top panels.

*Figure 7: These plots are very small. I think that showing differences without known the actual value is difficult to judge whether these anomalies are important or not.*

RE: We will increase the size of the plot, and as reviewer 1 suggests, stack them vertically as they all share the same x-axis.

*Figure 8: I do not understand this figure...* RE: Following the reorganization and shortening of the manuscript, we will remove this figure

*Figure 9: I also have difficulty understanding this figure, but I suspect that I am mostly confused by the phrasing in the caption: if the plots are showing the actual values, not the differences. What means "as a function of the model bias?"*

RE: Following the reorganization and shortening of the manuscript, we will remove this figure

Please also note the supplement to this comment:
https://tc.copernicus.org/preprints/tc-2020-52/tc-2020-52-AC2-supplement.pdf

[Figure]

**Supplement:**

**Revised outline of manuscript**

*In the following outline, we retained the figure numbering scheme of the manuscript.*

1. Introduction
2. Methods
   2.1. Study sites
      2.1.1. Utqiaġvik, Alaska
      2.1.2. Van Mijen Fjord
   2.2. Field measurements, sample and data processing
   2.3. Aggregation of ice core profile data sets
      2.3.1. Degree day classification
      2.3.2. Profile depth referenced to snow/ice and ice/ocean interface
   2.4. Model simulations
3. A spatial and temporal reference framework for analysis of ice core profile data sets
   3.1. Results
      New figure: Difference in classification by using DD vs date of coring or ice thickness for aggregating ice core data.
      Figure 3: Present ice core profile datasets with dual reference horizons (sea ice surface and bottom).
   3.2. Discussion: Advantages of the proposed framework
      We will discuss the benefits of using a DD framework and dual coordinate system referencing both the snow/ice and ice/ocean interface. We will discuss requirements in terms of sampling frequency, and amount of ice cores collected for each sampling event to produce a climatology (mean, variance, extremes of profile data) that meets the broader aims outlined in the paper.
4. Interannual and seasonal variability
   4.1. Results
      Figure 4: We will describe the shape of core profiles and their variability. We will highlight the difference between previous observations based on a single season (e.g., Eicken et al., 2002; Gough et al., 2012), and this long-term data set.
   4.2. Discussion
      4.2.1. Source of errors
         We will discuss sources of error (temperature bias, brine drainage), as well as inconsistencies or deviations from core sampling and processing protocols.
      4.2.2. Seasonal evolution and interannual variability
         We will discuss the environmental processes responsible for the observed spatial and seasonal variations in ice temperature and salinity.
5. Comparison with CICE model output
      Results
      Figure 5a, b: comparing modeled and observed ice and snow thickness to check the consistency of model tuning.
      Figure 7a, c, d: differences between model observation. This figure will be moved into supplemental materials.

Figure 6: Seasonal anomaly between model and observation: we will describe the success (absence of brine loss and bias in temperature measurement), and failure of the model (desalination during the melt season with more detail in Figure 5c).

Figure 5c: failure of model to capture desalination during the melt season

5.1. Discussion

5.1.1. Sources of error

We discuss the model calibration in terms of ice and snow thicknesses, and limitations of the stand-alone mode we used.

5.1.2. Seasonal evolution and interannual variability

As the model does not reproduce small-scale spatial variability, and does not capture errors due to the sampling method, we discuss the benefits  of the model in terms of reproducing seasonal evolution and interannual variability of salinity and temperature profiles and the implications for studies of sea ice processes.

6. Conclusions